



# How reliable are CMIP5 models in simulating dust optical depth?

Bing Pu[1,2] and Paul Ginoux[2]

[1]Atmospheric and Oceanic Sciences Program, Princeton University,

Princeton, New Jersey 08544

[2]NOAA Geophysical Fluid Dynamics Laboratory, Princeton, New Jersey 08540

*Correspondence to*: Bing Pu (bpu@princeton.edu)



**Abstract.** Dust aerosol plays an important role in the climate system by affecting the
radiative and energy balances. Biases in dust modeling may result in biases in simulating
global energy budget and regional climate. It is thus very important to understand how
well dust is simulated in the Coupled Model Intercomparison Project Phase 5 (CMIP5)
models. Here seven CMIP5 models using interactive dust emission schemes are
examined against satellite derived dust optical depth (DOD) during 2004-2016.

7         It is found that multi-model mean can largely capture the global spatial pattern

and zonal mean of DOD over land in present-day climatology in MAM and JJA. Global
mean land DOD is underestimated by -25.2% in MAM to -6.4% in DJF. While seasonal
cycle, magnitude, and spatial pattern are generally captured by multi-model mean over
major dust source regions such as North Africa and the Middle East, these variables are
not so well represented by most of the models in South Africa and Australia. Interannual
variations of DOD are neither captured by most of the models nor by multi-model mean.
Models also do not capture the observed connections between DOD and local controlling
factors such as surface wind speed, bareness, and precipitation. The constraints from
surface bareness are largely underestimated while the influences of surface wind and
precipitation are overestimated.

18        Projections of DOD change in the late half of the 21$^{st}$ century under the

Representative Concentration Pathways 8.5 scenario by multi-model mean is compared
with those projected by a regression model. Despite the uncertainties associated with both
projections, results show some similarities between the two, e.g., DOD pattern over
North Africa in DJF and JJA, an increase of DOD in the Arabian Peninsula in all seasons,
and a decrease over northern China from MAM to SON.





## 1. Introduction

Dust is the second most abundant aerosols by mass in the atmosphere after sea salt. It absorbs and scatters both shortwave and longwave radiation and thus modifies local radiative budget and consequently vertical temperature profile, influencing global and regional climate. For instance, studies found dust influences the strength of the West African monsoon  (e.g., Miller and Tegen, 1998; Miller et al., 2004; Mahowald et al., 2010; Strong et al., 2015) and Indian monsoonal rainfall (e.g., Vinoj et al., 2014; Jin et al., 2014, 2015, 2016;  Solmon et al., 2015; Kim et al., 2016; Sharma and Miller, 2017). Dust aerosols are also found to amplify droughts during the U.S. Dust Bowl and Medieval Climate Anomaly (Cook et al., 2008, 2009, 2013), and affect Atlantic tropical cyclones (e.g., Dunion and Velden, 2004; Wong and Dessler, 2005; Evan et al., 2006; Sun et al., 2008; Strong et al., 2018). Dust particles can also serve as ice cloud nuclei and influence the properties of the cloud (e.g., Levin et al., 1996; Rosenfield et al., 1997; Wurzler et al., 2000; Nakajima et al., 2001; Bangert et al., 2012) and affect regional radiative balance and hydrological cycle. When deposited in the oceans, iron-enriched dust also provides nutrients for phytoplankton, affecting ocean productivity and therefore carbon and nitrogen cycles and ocean albedo (e.g., Fung et al., 2000; Shao et al., 2011; Jickells et al., 2005).

Globally, the estimated radiative forcing from dust aerosol is 0.10 (-0.30 to +0.10) W m$^{-2}$, a magnitude about one fourth of the radiative forcing of sulfate aerosol or black carbon from fossil fuel and biofuel (Myhre et al., 2013; their Table 8.4). Biases in dust simulation may potentially affect global energy budgets and regional climate simulation.



Thus, it is very important to examine the capability of current state-of-the-art climate
models in simulating dust.
Only a few studies examined the Coupled Model Intercomparison Project Phase 5
(CMIP5) model output of dust and most of them are regional evaluations. For instance,
Evan et al. (2014) examined model output for Africa, but mainly focused on an area over
the northeastern Atlantic (10°–20°N and 20°–30°W) where a long-term proxy of dust
optical depth data over Cape Verde islands is available (Evan and Mukhopadhyay, 2010).
They found models underestimated dust emission and mass path and failed to capture the
interannual variations from 1960 to 2004, as models did not capture the negative
connection between dust mass path and precipitation over the Sahel.
Another work examined CMIP5 aerosol optical depth (AOD) is by Sanap et al.
(2014) for India. They compared dust distribution in the models with Earth Probe total
ozone monitoring system (EPTOMS)/ Ozone monitoring Instrument (OMI) aerosol index
(AI) from 2000 to 2005. They found most of CMIP5 models, except two HadGEM2
models, underestimated dust load over Indo-Gangetic Plains, and suggested the biases are
due to a misrepresentation of 850 hPa winds in the models. Later, Misra et al. (2016) also
examined CMIP5 modeled AOD for India but did not specifically focus on dust.
Shindell et al. (2013) examined the output of 10 models from the Atmospheric
Chemistry and Climate Model Intercomparison Project (ACCMIP) for one year (2000),
among which eight models also participated in the CMIP5. They noticed that simulated
dust AOD vary by more than a factor of two across models.  However, this study also did
not focus on dust, but emphasized the radiative forcings from anthropogenic aerosols.



None of the above studies examined global dust simulation in CMIP5 models.
What's more, most studies focused on annual mean, not seasonal averages. It is very
possible that models perform better in some seasons than others. AeroCom multiple-dust
model intercomparison was performed on both global and regional scales (Huneeus et al.,
2011) but only focused on one year, thus models' capability of simulating interannual or
long-term variability of dust is not clear. A comprehensive evaluation of the climatology
and interannual variation of global dust optical depth (DOD) in CMIP5 models will
provide a clear picture of model capability of dust simulation.
Here we examine the results of seven CMIP5 models (Table 1) by comparing
model output with DOD derived from Moderate Resolution Imaging Spectroradiometer
(MODIS) Deep Blue aerosol products. Projections on changes of DOD in the late half of
the 21$^{st}$ century by CMIP5 models and also by a regression model (Pu and Ginoux, 2017)
are examined and analyzed.
The following section introduces data and methods used in this study. Results are
presented in section 3, including examinations on the climatology and interannual
variations of modeled DOD and future projections. Discussion and major conclusions are
presented in sections 4 and 5, respectively.

**2. Data and Methodology**
**2.1 DOD from MODIS**
DOD is a widely used variable that describes optical depth due to the extinction
by mineral particles. It is one of the key factors (single scattering albedo and asymmetry
factor being the two others) controlling dust interaction with radiation. Monthly DOD are



derived from MODIS aerosol products retrieved using the Deep Blue (MDB2) algorithm,
which employs radiance from the blue channels to detect aerosols globally over land even
over bright surfaces, such as desert (Hsu et al., 2004, 2006). Ginoux et al. (2012b) used
collection 5.1 level 2 aerosol products from MODIS aboard the Aqua satellite to derive
DOD. Here, both MODIS aerosol products (collection 6, level 2; Levy et al., 2013) from
the Aqua and Terra platforms are used. Aerosol products such as AOD, single scattering
albedo, and the Ångström exponent are first interpolated to a regular 0.1° by 0.1° grid
using the algorithm described by Ginoux et al. (2010). The DOD is then derived from
AOD following the methods of Ginoux et al. (2012b) with adaptions for the newly
released MODIS collection 6 aerosol products (Pu and Ginoux, 2016).
Daily DOD is derived for both Aqua and Terra satellites and then averaged to
monthly data and interpolated to a 1° by 1° grid. Terra passes the Equator from north to
south around 10:30 local time while Aqua passes the Equator from south to north around
13:30 local time. To reduce missing data and also to combine the information from both
morning and afternoon hours, a combined monthly DOD (here after MODIS DOD) is
derived by averaging Aqua and Terra DOD when both products exist or using either
Aqua or Terra DOD when only one product is available. As shown in Figure S1 in the
Supplement, the mean available days in each season and also spatial coverage are
enhanced in combined DOD than using Aqua or Terra (not shown) DOD alone. This
combined DOD is available from January 2003 to December 2016.
Aqua and Terra DOD product has previously been used to study global dust
sources (Ginoux et al., 2012b), dust variations in the Middle East (Pu and Ginoux, 2016)
and the U.S. (Pu and Ginoux, 2017), and has been validated with Aerosol Robotic





NETwork (AERONET) stations over the U.S. (Pu and Ginoux, 2017). Here we compared
MODIS DOD climatology with both AERONET observation and DOD retrieved from
Cloud-Aerosol Lidar with Orthogonal Polarization (CALIOP; Winker et al., 2004; 2007)
aboard the Cloud-Aerosol Lidar and Infrared Pathfinder Satellite Observation
(CALIPSO) satellite. AERONET stations provide cloud-screened and quality assured
(level 2) coarse mode aerosol optical depth (COD) at 500 nm, which is processed by the
Spectral Deconvolution Algorithm (O'Neill et al., 2003). Only nine sites have COD
records during 2003-2016, and the climatological mean of MODIS DOD generally
compares well with these sites (Figure S2 in the Supplement).

CALIOP measures backscattered radiances attenuated by the presence of aerosols

and clouds and retrieves corresponding microphysical and optical properties of aerosols.
Monthly dust AOD (or DOD) on a 2° latitude by 5° longitude grid are available since
June 2006. The climatology of CALIOP DOD during 2007-2016 is similar to that of
MODIS DOD during the same period (Figure S3 in the Supplement). The global mean
(over land) MODIS DOD is slightly higher than that from CALIOP, probably due to the
lower horizontal resolution of the latter. The pattern correlations (e.g., Pu et al., 2016)
between the two products range from 0.83 in boreal spring and summer to 0.63 in boreal
winter (Figure S3 in the Supplement).

Due to higher spatial resolution (compared with CALIOP) and coverage

(compared with AERONET sites), MODIS DOD is chosen as the primary product to
validate CMIP5 model output. Nine regions (Table 2) are selected to study the DOD
magnitude, spatial pattern, and variations. These regions cover major dust source regions
previously identified (Ginoux et al. 2012).



## 2.2 Reanalysis and observation datasets


To examine the interannual variations of DOD and its connection with local
controlling factors such as surface wind speed, bareness, and precipitation, monthly data
of 10 m wind speed from the ERA-Interim (Dee et al., 2011), leaf area index (LAI) data
from Advanced Very High Resolution Radiometer (AVHRR; Claverie et al., 2014,
2016), and precipitation from the Precipitation Reconstruction over Land (PRECL; Chen
et al., 2002) are used.
ERA-Interim is a global reanalysis from the European Centre for Medium-Range
Weather Forecasts (ECMWF). Its horizontal resolution is T255 (about 0.75° or 80 km),
very suitable to study the influence of wind speed on dust emission and transport on
small scales. The monthly data are available from 1979 to present day.
Monthly LAI derived from the version 4 of Climate Data Record (CDR) of
AVHRR is used to calculate surface bareness.  The data are produced by the National
Aeronautics and Space Administration (NASA) Goddard Space Flight Center (GSFC)
and the University of Maryland. Monthly gridded data on a horizontal resolution of 0.05°
by 0.05° degree are available from 1981 to present. This product is selected due to its
high spatial resolution and long temporal coverage. Surface bareness is calculated from
seasonal mean LAI as the following,
$$Bareness = exp\ (-1 \times LAI) \qquad . \qquad (1)$$
PRECL precipitation from the National Oceanic and Atmospheric Administration
(NOAA) is a global analysis available monthly from 1948 to present at a 1° by 1°
resolution. The dataset is derived from gauge observations from the Global Historical
Climatology Network (GHCN), version 2, and the Climate Anomaly Monitoring System





(CAMS) datasets. Its long coverage and relatively high spatial resolution is quite suitable
to study the connections between DOD and precipitation.

**2.3 CMIP5 model output**

Among CMIP5 models we selected seven models that used interactive dust

emission schemes, in which dust emission varied in response to changes of climate. The
output of 10 m wind speed, precipitation, and LAI are also available from these models.
Other models (to our best knowledge) either used offline dust as an input, in which dust
emission did not interactively respond to meteorological and climate changes, or did not
write out the variables needed for this analysis.

Both historical run from 1861 to 2005 and future run under the Representative

Concentration Pathways 8.5 (RCP 8.5) scenario (Riahi et al., 2011) from 2006 to 2100
are used. Here the RCP 8.5 scenario is chosen because it represents the upper limit of the
projected greenhouse gas change in the twenty-first century and thus likely is the worst-
case scenario for future DOD variation under climate change. Also, studies found that
observed $CO_2$ emission pathway during 2005-2014 matches RCP 8.5 scenario better than
other scenarios (e.g., Fuss et al., 2014), which makes the RCP8.5 output suitable to
examine present-day DOD variations after 2005.

Monthly model output of dust load, surface 10 m wind speed, precipitation, and

LAI are used. Historical output from 2003 to 2005 and RCP 8.5 output from 2006 to
2016 are combined to form time series and climatology during 2003-2016 to compare
with MODIS DOD during the same time period.



### 183   2.3.1 DOD derived from modeled dust load

Most CMIP5 models did not save DOD, so we used monthly dust load and
converted them to DOD using the relationship derived by Ginoux et al. (2012a) as the
following
$$\tau = M \times e \ ,\tag{2}$$
where $\tau$ is DOD at 500 nm, $M$ is the load of dust in unit of (g m$^{-2}$), and $e = 0.6$ m$^2$ g$^{-1}$ is
the mass extinction efficiency. Dust load from different models is first interpolated to a
2° by 2.5° grid and then converted to DOD. The same method was used by Pu and
Ginoux (2017) for the U.S. We compared the derived DOD with modeled DOD from one
historical simulation of GFLD-CM3 model (Donner et al., 2011). The pattern correlation
of the climatology (1861-2005) between the derived DOD and modeled DOD are very
high, all above 0.99 for four seasons (not shown).  The percentage differences between
derived DOD and modeled DOD averaged over global land range from -3.6% in DJF and
SON to 1.3% in MAM and JJA. Over Africa, DOD is slightly overestimated by 0~6.7%
(regional mean), while over the Middle East, there is a small underestimation by -1.6% in
SON and up to 8.2% overestimation in JJA. Among the nine regions we focused in this
analysis, three regions (North America, South Africa, and South America) show an
underestimation of more than 20% in some seasons and two regions (Northern China and
Australia) show an overestimation of more than 10% in some seasons.

### 203   2.4 Multiple linear regression

In order to examine the relative contribution of each local controlling factor to
DOD variations, multiple linear regression is applied by regressing DOD onto





standardized seasonal mean ERA-Interim surface wind speed, AVHRR bareness, and
PRECL precipitation at each grid point. All the data are re-gridded to a 1° by 1° grid
before the calculation. Over regions where values are missing for any of the explanatory
variables (i.e., precipitation, bareness, and surface wind speed) or DOD, the regression
coefficients are set to missing values. The collinearity among these explanatory variables
is examined by calculating variance inflation factor (VIF) (e.g., O'Brien, 2007; Abudu et
al., 2011), and in most regions the VIF is below 2 (not shown), indicating a low
collinearity (5–10 is usually considered high). Bootstrap resampling is used to test the
significance of the regression coefficients, following the method used by Pu and Ginoux

(2017).

Multiple linear regression is also applied to CMIP5 model derived DOD and

output of surface wind speed, bareness, and precipitation to obtain regression coefficients
from the models.  All variables are interpolated to a 2° by 2.5° grid before regression.
The results are compared with regression coefficients derived from observational
datasets.

Similar to the method used by Pu and Ginoux (2017), the regression coefficients

derived from MODIS DOD and observed controlling factors from 2004 to 2016 and
CMIP5 model output of surface wind speed, bareness, and precipitation are used to
project variations of future DOD. Here we tried two groups of CMIP5 output for these
controlling factors. One group used seven models with interactive dust emission scheme
(Table 1), and the other used 16 CMIP5 models as did by Pu and Ginoux (2017; their
Supplementary Table S1). The reason to test the latter is to include as much model output
of the controlling factors as possible. The differences between the historical run (1861–



2005 average) and that of the RCP 8.5 run for the late half of the twenty-first century
(2051–2100) are standardized by the standard deviation of the historical run for each
explanatory variable. The projected change reveals how DOD will vary with reference to
the historical conditions (mean and standard deviation).

**3. Results**
**3.1 Climatology (2004-2016)**

Figure 1 shows the climatology of MODIS DOD (top panel) in four seasons

during 2004-2016 and that from the CMIP5 multi-model mean (bottom). Globally, the
dustiest regions are largely located over the northern hemisphere (NH) over North Africa,
the Middle East, and East Asia (Figs. 1a-d). In these regions, DOD is higher in boreal
spring and summer than fall and winter. Modeled global DOD over land is generally
lower than that from MODIS DOD, ranging from -0.028 (-25.2%) in MAM to -0.005 (-
6.4%) in DJF. The global spatial pattern is better captured in MAM and JJA, with pattern
correlations of 0.74 and 0.85, respectively (Figs. 1f-g). In DJF, DOD is overestimated
over central Africa and Australia, but underestimated over the Middle East and Asia (Fig.
1e), while in SON there is a similar overestimation in Australia and an underestimation in
the Middle East (Fig. 1h).

Figure 2 shows the zonal mean of CMIP5 DOD from individual models (thin

colorful lines) and multi-model ensemble mean (thick black), in comparison with MODIS
DOD (thick red). In DJF, DOD is underestimated in the NH from 15° N to 50°N but
overestimated over the tropics and southern hemisphere (SH) (Fig. 2a). While the
overestimation in the SH is largely contributed by three models, the underestimation in



the NH appears in all the seven models. The overestimation of DOD in HadGEM2-ES
has also been identified in a previous study (Bellouin et al., 2011) and will be discussed
later. In MAM, a similar overestimation of DOD in the tropics and SH also occurs in
some models, and the multi-model mean slightly overestimates DOD around 20°-30°S
(Fig. 2b). In NH, there is a weak underestimation too, but the overall gradient is largely
captured. In JJA, the multi-model mean resembles MODIS DOD very well (Fig. 2c),
consistent with the highest pattern correlation in this season shown in Fig. 1. The peak
around 19° N in North Africa and Middle East is well captured by the multi-model mean,
although the magnitude is slightly underestimated. In SON, different from MODIS DOD
that peaks around 19°N, the multi-model mean has two peaks around 15°N and 28°S,
respectively, a pattern somewhat similar to that in DJF (Fig. 2d). Consequently, DOD in
CMIP5 multi-model mean is overestimated at 15°-40°S and 0°-15°N but underestimated
at 15°S -0° and 15°-40°N.
Seasonal cycles of CMIP5 DOD are compared with MODIS DOD in nine regions
in Figure 3. The annual means of DOD in each region from multi-model mean (black)
and MODIS (red) are also listed in each plot. The spread of DOD among individual
models is greater during boreal spring and summer for regions in the NH and during
austral spring and summer for regions in the SH than other seasons. Seasonal cycles over
North Africa, the Middle East, North America and India are generally captured, with
modeled DOD peaking during the same seasons as MODIS DOD. Over northern China,
MODIS DOD peaks in spring, consistent with previous studies (e.g., Zhao et al., 2006;
Laurent et al., 2006; Ginoux et al., 2012b), while multi-model mean peaks much later in
June. Similar misrepresentation occurs over the southeastern Asia. In South Africa and



South America the observed maxima in early austral spring (i.e., September) are also
missed. In Australia, DOD is largely overestimated and the peak from November to
January in MODIS DOD is also misrepresented in the multi-model mean. Similar to the
finding here, Bellouin et al. (2011) also found that HadGEM2-ES model overestimated
DOD over Australia and Thar desert region in northwestern India and suggested that
these overestimations were likely due to model's overestimation of bare soil fraction and
underestimation of soil moisture.

We further examine the magnitudes and spatial patterns of CMIP5 DOD in these

regions. Figure 4 shows the ratio of pattern standard deviations (standard deviations of
values within the domain) and pattern correlation between CMIP5 DOD and MODIS
DOD climatology (2004-2016) in each region for four seasons. While the former reveals
the magnitude differences, the latter demonstrates the spatial resemblance.

Over North Africa, the Middle East, and India, the ratio of CMIP5 DOD from

individual models and multi-model mean versus MODIS DOD are all within ± one order
of magnitude (Fig. 4). Most models underestimate DOD in northern China, although the
magnitudes are largely within the range of -one order of magnitude to one. Over North
America, South Africa, and Australia, some models underestimate the DOD by more than
two orders of magnitudes, while over Australia three models overestimate DOD by more
than one order of magnitude. In general, magnitudes of multi-model mean are closer to
satellite DOD than most individual models and are largely within ± one order of
magnitude of MODIS DOD.

The spatial patterns are better captured over North Africa and the Middle East

than other regions (Fig. 4), with pattern correlations above 0.6 in most models (with



highest pattern correlation of 0.92 and 0.83, respectively). Pattern correlations from
multi-model mean are also high, reaching 0.87 (0.78) over North Africa and 0.75 (0.73)
over the Middle East in JJA (MAM). Nonetheless, some models show negative pattern
correlations over North Africa, northern China, North America, southeastern Asia, South
Africa, South America, and Australia. Overall, spatial patterns are less well represented
in regions over the SH than over the NH in CMIP5 models.

In short, in terms of both magnitudes and spatial pattern, DOD climatology is best

represented over North Africa and the Middle East among the nine regions.  The multi-
model mean shows that DOD over North Africa is slightly better simulated than over the
Middle East, somewhat similar to the finding of AeroCom multi-model analysis
(Huneeus et al. 2011).

**3.2 Interannual variations**

An important aspect of dust activity is its long-term variations, including

interannual and decadal variations. Dust emission in North Africa is known to have
strong decadal variations (e.g., Prospero and Nees, 1986; Prospero and Lamb, 2003;
Mahowald et al., 2010; Evan et al., 2014, 2016), while over Australia, strong interannual
variations have been related to El Niño–Southern Oscillation (e.g., Marx et al., 2009;
Evans et al., 2016). Due to the short time coverage of high quality satellite products, we
focus on interannual variations of DOD from 2004 to 2016.

Figure 5 shows the correlations of regional mean time series of DOD between

MODIS and CMIP5 models and multi-model mean for each season in nine regions. We
also show correlations between the reconstructed DOD and MODIS DOD for reference



(Table S1 in the Supplement). Previous study found that the variations of dust event
frequency over the U.S. in the recent decade could be largely represented by the
variations of three local controlling factors: seasonal mean surface wind speed, bareness,
and precipitation (Pu and Ginoux, 2017). These factors have previously been found to
constrain dust emission or variability on multiple time scales (e.g., Gillette and Passi,
1988; Fecan et al., 1999; Zender and Kwon, 2005). While surface wind is positively
related to the emission and transport of dust, vegetation is an important non-erodible
element that prevents soil erosion from wind. Precipitation is generally negatively related
to dust emission and transport processes. While the scavenging effect of precipitation on
small dust particles only lasts a few hours or days, influences of precipitation on soil
moisture lasts longer. Here we extend our regression model (Pu and Ginoux, 2017) to a
global scale. Regression coefficients are obtained by regressing MODIS DOD onto
observed surface wind, bareness, and precipitation during 2004-2016 (see methodology
section for details). The reconstructed DOD is then calculated using these regression
coefficients and time-varying controlling factors.

The interannual variations of DOD are in general not well captured by CMIP5

models. This is consistent with previous study by Evan et al. (2014) who found dust
variability downwind of North Africa over the northeastern Atlantic was misrepresented
in CMIP5 models. In most regions, only one or two models show significant positive
correlation with MODIS DOD in some seasons, and negative correlations exist in all
regions (Fig. 5). North Africa, the Middle East, southeastern Asia, South America, and
Australia show less negative correlations than other dusty regions. On the other hand,
reconstructed DOD shows significant positive correlations with MODIS DOD over most





regions in all seasons (Table S1 in the Supplement). This suggests that the interannual
variations of DOD can be largely attributed to the variations of these controlling factors,
and models probably misrepresented these relationships, in addition to their incapacity of
capturing the interannual variations of individual controlling factors in general (Figures
S4-6 in the Supplement), which is not uncommon for coupled models.

We further examine the connection between those controlling factors and DOD in

CMIP5 models. Figure 6 shows the dominant controlling factors among the three (surface
wind speed, bareness, and precipitation) on DOD variations in four seasons from MODIS
(left column) and from CMIP5 multi-model mean (right column), respectively. To
highlight factors controlling DOD variations near the dust source regions, a mask of
AVHRR LAI≤ 0.5 is applied to both coefficients.

Bareness plays the most important role in many dusty regions in observations,

e.g., over Australia, central U.S., and South America (Figs. 6a-d). Note that while
bareness plays an important role over the Sahel during DJF and MAM, it also shows
strong signal over some areas in the northern North Africa (Figs. 6a-b). The reliability of
this information is limited by the accuracy of LAI retrieval in these areas. The value of
bareness in this region is actually quite high (as LAI is very low), but still has weak
interannual variability (Figures S7 in the Supplement). Over some areas of North and
South Africa, the Middle East, and East Asia, surface wind and precipitation are also
quite important.

The role of bareness is largely underestimated in CMIP5 models, while surface

wind and precipitation become the dominant factors (Figs. 6e-h). The misrepresentation
of the connection between DOD and these controlling factors may cause the



misrepresentation of the dust load and its variability. Taking Australia for an example,
the overestimation of DOD magnitudes may be related to an overestimation of the
influence of surface wind on DOD and a lack of constraints from surface bareness.

Despite the large differences between the observed and modeled connections

between DOD and the controlling factors, some regions show similarities. For instance,
over North Africa in DJF, both show an important influence from surface winds (Figs.
6a, e), although the locations of surface wind-dominant areas are not exactly the same.
Evan et al. (2016) also found a dominant role of surface wind on African dust variability,
but they focused on monthly means, not seasonal averages. In MAM, precipitation starts
to play a role in some parts of North Africa, while surface wind still dominates in some
areas (Fig. 6b). Same increasing influence of precipitation is shown in the multi-model
mean, but such an influence seems overestimated (Fig. 6f).  In JJA, the influences of
surface wind in North Africa and precipitation and bareness in the Middle East in the
multi-model mean (Fig. 6g) also show some similarity to observation (Fig. 6c), although
an underestimation of the influence from bareness and an overestimation of surface wind
are still there.

Also, note that in CMIP5 models, due to lack of constraints from low surface

temperature (e.g., over frozen land) and snow cover on dust emission or
misrepresentations of dust transport, DOD and also the regression coefficients still exist
over NH high latitudes in boreal winter and spring in the multi-model mean (Figs. 6e-f).







## 3.3 Future projections

How will DOD change in response to increasing greenhouse gases? The results from CMIP5 multi-model mean are shown in Figure 7. We compare the DOD during the late half of the 21$^{st}$ century under the RCP 8.5 scenario with that in the historical level (1861-2005 average).

Over land, CMIP5 model projects a decrease of global mean DOD in all seasons except JJA (Figs. 7a-d). The inter-model standard deviation is much greater than the multi-model mean, suggesting large discrepancies among individual models. The projected decrease is largely over northern North America, southern North Africa, eastern central Africa, and East Asia, while the increase is largely over northern North Africa, the Middle East, southern North America, South Africa, South America, and southern Australia (Fig. 7).  Regional means of DOD change (in percentage) with reference to CMIP5 historical run are summarized in Table 3.

What might be the causes of DOD change? Figure 8 shows the projected change of precipitation, bareness, and surface wind speed from CMIP5 multi-model mean. These factors play important role in DOD variations in the present day, although models tend to underestimate the role of bareness and overestimate the influences of precipitation and surface wind (Fig. 6). Increases in precipitation can increase soil moisture and remove airborne dust, thus usually favors a decrease of DOD.  As shown in Figs. 8a-d, the increases of precipitation in northern Eurasia, northern North America, the Congo basin in Africa, and Australia (DJF and MAM) may contribute to the decrease of DOD in these regions, while the decreases of precipitation over northern North Africa and the Middle East (DJF and MAM), South Africa, and South America may contribute to the increase of



413 DOD (DJF-SON). Also note that in JJA both precipitation and DOD increase over

414 northern North Africa and the Middle East (Fig. 8c), suggesting other factors dominate

415 the variation of DOD in the multi-model mean.

416  A decrease (increase) of bareness indicates a growth (decay) of vegetation and is

417 usually associated with a decrease (increase) of DOD. In general, except regions such as

418 southern North America, South America, South Africa, part of northern Eurasia, and

419 central Sahel, the pattern of bareness change does not resemble DOD change (Figs. 8e-h).

420 This is probably due to the fact that the overall influence of bareness on DOD variation is

421 underestimated in CMIP5 models (Fig. 6).

422  Increases in surface wind can enhance dust emission and transport, and vise versa.

423 The changes of surface wind in DJF and MAM are similar and likely to contribute to the

424 increase of DOD over northern North Africa, the Middle East, eastern South America,

425 southern South Africa, southern Australia (Figs. 8i-j). The decrease of DOD over

426 northwestern North America, the Sahel, and northern Australia may also relate to the

427 decrease of surface wind there, in addition to an increase of precipitation and a reduction

428 of bareness. In JJA and SON (Figs. 8k-l), the increases of surface wind in South America,

429 South Africa, central Australia and the decreases of wind in northwestern North America,

430 northern Eurasia, and the central Sahel are also consistent with patterns of DOD change.

431  In short, variations of CMIP5 DOD in the late half of the 21st centaury are more

432 consistent with changes of precipitation and surface wind speed than with surface

433 bareness, consistent with the analysis above regarding to the present-day condition.

434  The projected change of DOD from the regression model is shown Figure 9. The

435 results are calculated using the regression coefficients obtained from observations during



2004-2016 and projected changes of precipitation, bareness, and surface wind speed from
16 CMIP5 models (see methodology). A similar method is applied to the model output
from seven CMIP5 models with interactive dust emission scheme, and results are similar
(Figure S8 in the Supplement) A mask of present-day LAI ≤ 0.5 is also applied to
highlight the changes of DOD near dust source regions. By doing this, we assume the
location of major dust sources will not change much at the late half of the 21$^{st}$ century.
The unmasked figure is presented in the supplementary file (Figure S9 in the
Supplement). The reason we did not use the projected future LAI as a mask is that
there're large uncertainties associated with LAI projection, especially over northern
hemisphere subtropical regions (e.g., Figs. 8e-h).

In DJF, regression model projected change of DOD over Mexico, North Africa,

the Middle East and part of northern China (Fig. 9a) are similar to those projected by
CMIP5 models over those dust source regions (Fig. 7a), but with a greater magnitude. In
MAM, a decrease of DOD is projected over large area of North Africa (Fig. 9b), which is
different from the pattern projected from the CMIP5 multi-model mean (Fig. 7b). The
decrease of DOD over northern central U.S. is also different from the overall increase
projected by CMIP5 DOD, as also noted by Pu and Ginoux (2017). However, the
increase of DOD over the Middle East and the decrease of DOD over northern China are
similar to that of CMIP5 DOD. During JJA and SON, DOD decreases over the Sahel and
northern China but increases over a belt to the north of central Sahel and parts of the
Middle East (Figs. 9c-d).  The weak increase of DOD over the southern corner of South
Africa in JJA and a slight decrease in SON also has high agreement among the models.



Changes of DOD over Australia are very small and show little consistency among the
models.

The contribution of each controlling factor to the total DOD change is shown in

Figure 10. While changes of bareness over North Africa, northern Middle East and
northern China play an important role in DOD change, changes of precipitation, e.g. over
northwestern China in MAM, and surface wind, e.g., over North Africa and the Middle
East in DJF and MAM, also play vital roles.

Both projections from the CMIP5 models and that from the regression model have

large uncertainties. The reliability of future projection by CMIP5 models is limited by
models' capability of capturing present-day climatology and observed connection
between DOD and local controlling factors. As discussed earlier, the overall performance
of models is better in those very dusty regions in the NH, such as North Africa and the
Middle East, than other regions. Multi-model mean also overestimates the connection
between DOD and precipitation and surface wind and underestimates the influence of
bareness, which can cast doubts on the projected variation of DOD in response to climate
change.

The uncertainties associated with regression model are two folds. First, there're

uncertainties associated with the regression model itself. Since the regression coefficients
are derived from observed relationships between DOD and controlling factors in a
relatively short time period, factors controlling the low frequency variation of DOD (e.g.,
decadal variations) may not be included. Other meteorological factors that could play an
important role in regional dust variability, e.g., nocturnal low-level jets (e.g., Todd et al.,
2008; Fiedler et al., 2013; Fiedler et al., 2016) and haboobs over Africa (e.g., Ashpole



and Washington, 2013), are not directly considered in the model. The influences of
anthropogenic land use/land cover change are also not included in the regression model.
Anthropogenic land use/land cover change has been found to have played an important
role in long-term dust variability in some regions (e.g., Neff et al., 2005; 2008; Moulin
and Chiapello, 2006; McConnell et al., 2007), although previous modeling study found
its influences on future dust emission was minor compared to climate change (Tegen et
al., 2004). So the projection made by the regression model only reveals the change of
DOD in association with climate change. Second, uncertainties associated with model
projected change of controlling factors, such as bareness in U.S. in JJA as pointed by Pu
and Ginoux (2017), also limit the accuracy of the results.
Despite these uncertainties, both methods make similar projections particularly in
some dusty regions. For instance, the DOD pattern over North Africa in DJF and JJA, an
increase of DOD in the Arabian Peninsula in all seasons, and a decrease of DOD over
northern China from MAM to SON (Figs. 7, 9).

**4. Discussion**
We examined DOD in seven CMIP5 models with interactive dust emission
schemes. Other important variables that influence the radiative property and
concentration of dust, such as Angström exponent, dust emission, and surface
concentration, are worth further examination, if these variables are archived.  A better
quantification of the radiative forcing of dust may also require an examination on the size
distribution of dust particles, as studies (e.g., Kok et al., 2017) found in current AeroCom



models fraction of coarse dust particles were underestimated and so was the warming
effect of dust. Whether this is the case in the CMIP5 models is not clear.

Early studies on future dust projection used offline dust models driven by climate

model output under different scenarios. For instance, Mahowald and Luo (2003) used an
offline dust model and output from National Center of Atmospheric Research's coupled
Climate System Model (CSM) 1.0 (Boville and Gent, 1998) under A1 scenario
(Houghton et al., 2001) and projected a decrease of dust emission by the end of the $21^{st}$
century by -20% to -63%, depending on different scenarios. In general, when they
included vegetation change, the projected dust reduction became greater, but including
land use change slightly weakened such reduction. Similarly, Tegen et al. (2004) used
output from ECHAM4 and HadCM3 and a dust model (Tegen et al., 2002) to examine
the change of dust emission by 2040-2050 and 2070-2080 and found results were model
and scenario dependent, from -26% to 10%. However, including anthropogenic
cultivation practices tended to increase dust emission in both models. They also pointed
out that such an influence from anthropogenic land-use was not big enough to overcome
the effect of climate change.

The interactive dust emission schemes and new generations of climate models

used in CMIP5 are likely to provide more reliable projections, but this may also depend
on how changes of dust and its radiative forcing are fed back to the climate system in the
models. While these projections are largely model-dependent, based on our analysis on
the DOD climatology in CMIP5 models, the multi-model mean has a better chance to
provide a more reliable projection than individual models.


Here a regression model combined with MODIS DOD is used to identify key
local factors that control the variation of DOD on the interannual time scale. The results
are then compared with model output to examine models' capability of capturing
observed connections between DOD and controlling factors. This method may be applied
to other dust model intercomparison projects as well, such as AeroCom (Huneeus et al.
2011), to help examine model performance.

**5. Conclusion**
Dust aerosol plays an important role in the climate system by directly scattering
and absorbing solar and longwave radiation and indirectly affecting the formation and
radiative properties of cloud. It is thus very important to understand how well dust is
simulated in the state-of-the-art climate models. While many features and variables are
systematically examined in the CMIP5 multi-model output, we found that to our best
knowledge an evaluation of global dust modeling in CMIP5 models is still in blank. In
this study we examined a key variable associated with dust radiative effect, dust optical
depth (DOD), using seven CMIP5 models with interactive dust emission schemes and
DOD retrieved from MODIS Deep Blue aerosol products.
We found that the global spatial pattern and magnitude are largely captured by
CMIP5 models in the 2004-2016 climatology, with an underestimation of global DOD
(over land) by -25.2% in MAM to -6.4% in DJF. The spatial pattern is better captured in
boreal dusty seasons during MAM and JJA. In JJA, the simulated zonal mean DOD from
multi-model mean captures MODIS DOD quite well.





The magnitudes of multi-model mean are closer to MODIS climatology than most
individual models and are largely within ± one order of magnitude of MODIS DOD in
the nine regions examined here (North Africa, the Middle East, Northern China, North
America, India, southeastern Asia, South Africa, South America, and Australia; see Fig. 1
and Table 2 for domains). While some models underestimate DOD in North America and
South America by more than two orders of magnitude, a few also overestimate DOD in
Australia by more than one order of magnitude. Both the magnitude and spatial patterns
of DOD are better captured over North Africa and the Middle East than other regions.
The multi-model mean also largely captures the seasonal cycle of DOD in some
very dusty regions, such as North Africa and the Middle East. Seasonal variations in
North America and India are also generally captured, with the modeled DOD peaking at
approximately the same season as in MODIS DOD, but not so in Northern China and
southeastern Asia. Seasonal cycles in those dusty regions in the southern hemisphere is
generally not well captured, with modeled DOD over South Africa and South America
peaking later than that in MODIS DOD but earlier in Australia.
The interannual variations of DOD are not captured by most of the CMIP5
models during 2004-2016. This is likely due to models' underestimation of the
constraints from surface bareness on dust and overestimation of the influences from
surface wind speed and precipitation in those major dust source regions, in addition to the
fact that coupled models usually do not capture the observed interannual variations of
precipitation, surface wind, and bareness as well. CMIP5 model projected change of
DOD in the late half of the 21$^{st}$ century (under the RCP 8.5 scenario) with reference to
historical condition (1861-2005) also shows greater influence from precipitation and



surface wind change than from surface bareness. Overall, multi-model mean projects a
change of DOD over land from -3.8% in SON to 3.3% in JJA.

We also provide a projection of future DOD change using a regression model

based on local controlling factors such as surface wind, bareness, and precipitation (Pu
and Ginoux, 2017). This model can largely capture the interannual variations of DOD in
2004-2016. The regression model projects a reduction of DOD in the Sahel in all seasons
in the late half of the 21$^{st}$ century under the RCP 8.5 scenario, largely due to a decrease of
surface bareness. DOD is projected to increase over the southern edge of Sahara in
association with surface wind and precipitation changes except in MAM, when a
reduction of DOD over most part of North Africa is projected.  DOD is also projected to
increase over the Arabian Peninsula in all seasons and to decrease over northern China
from MAM to SON.

Despite large uncertainties associated with both projections, we find some

similarities between the two, which may be informative, for instance, changes of DOD
over North Africa in DJF and JJA, an increase of DOD in the Arabian Peninsula in all
seasons, and a decrease of DOD over northern China from MAM to SON.











*Acknowledgements.*

This research is supported by NOAA and Princeton University's Cooperative

Institute for Climate Science and NASA under grant NNH14ZDA001N-ACMAP. The
authors thank Drs. Songmiao Fan and Fabien Paulot for their helpful comments on the
early version of this paper.

PRECL Precipitation data are provided by the NOAA/OAR/ESRL PSD, Boulder,

Colorado, USA, from their web site at *http://www.esrl.noaa.gov/psd/*. The  CALIOP
products are downloaded from *https://www-*
*calipso.larc.nasa.gov/tools/data_avail/dpo_read.php?y=2007&m=08&d=10*. AVHRR
leaf area index data are available at: *ftp://eclipse.ncdc.noaa.gov/pub/cdr/lai-fapar/files/*.
The ERA-Interim is downloaded from http://www.ecmwf.int/en/research/climate-
reanalysis/era-interim. The AERONET coarse mode aerosol optical depth data are
downloaded from https://aeronet.gsfc.nasa.gov/. CMIP5 data are downloaded from:
https://pcmdi.llnl.gov/projects/esgf-llnl/.











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





Table 1 CMIP5 models used in this study. Models tagged with plus signs (+) considered
anthropogenic land use/land cover change in their vegetation prediction.

Table 2 List of regions selected to compare model output with MODIS DOD. Locations
of these regions are also plotted in Fig. 1b. Acronyms are used for some regions for short,
and are listed in the brackets in the first column. Note that the region names such as
Northern China and India are not exactly the same as their geographical definitions but
also covers some areas from nearby countries.

Table 3 Changes of DOD in the late half of the 21$^{st}$ century (2051-2100; RCP 8.5
scenario) from the historical condition (1861-2005) projected by CMIP5 multi-model
mean (second to fifth columns) and the regression model (sixth to ninth columns) in the
nine regions. Changes of DOD are shown in percentage with reference to CMIP5 multi-
model historical run. Note that in some regions the projected change by the regression
model is quite large (i.e., greater than ± 100%), largely due to the underestimation of
CMIP5 historical run in these regions.









Figure 1.  Figure 1.  Climatology (2004-2016) of Aqua and Terra combined DOD (i.e.,
MODIS DOD; top panel) and multi-model mean of CMIP5 DOD (bottom) for four
seasons. The pattern correlation (centered; calculated after interpolating MOIDS DOD to
CMIP5 DOD grids) between CMIP5 and MODIS DOD are shown in pink in the bottom
panel. Blue numbers denote global mean DOD over land. For CMIP5 model results, ±
one standard deviation among seven CMIP5 models is also shown. Black boxes in (b)
denote nine averaging regions (Table 2). Here we only added these boxes in (b) instead of
every plot to keep the figure clean.  Note that CMIP5 multi-model mean is masked by
MODIS DOD for comparison. Dotted area in (e)-(h) shows where multi-model mean is
greater than one inter-model standard deviation.

Figure 2. Zonal mean DOD from MODIS (thick red), CMIP5 multi-model mean (thick
black), and each individual model (other colorful lines).

Figure 3. Seasonal cycle of DOD in nine regions (Table 2) averaged over 2004-2016.
Thick red lines denote MODIS DOD, thick black lines denote CMIP5 multi-model mean,
and other colorful lines denote individual model output. The annual means from MODIS
DOD (Obs; red) and multi-model mean (Ens; black) are shown  in each panel.

Figure 4. Spatial statistics comparing DOD from CMIP5 models with that from MODIS
in nine regions. Label on the X-axis shows individual models (1-7) and multi-model
mean (8). Y-axis shows the ratio of pattern standard deviations between model
climatology (2004-2016) and that of MODIS, which reveals the relative amplitude of the



simulated DOD versus satellite DOD. The color denotes pattern correlation (centered)
between each model and MODIS DOD in each region.

Figure 5. Correlations (color) between regional averaged time series from CMIP5 DOD
and MODIS DOD from 2004 to 2016 for four seasons. Numbers in the X-axis denotes
each model (1-7) and multi-model mean (8). Correlations significant at the 90%
confidence level are marked by a star and significance at the 95% confidence level by
two stars.

Figure 6. Regression coefficients calculated by regressing DOD in each season onto
standardized precipitation (purple), bareness (orange), and surface wind speed (green)
from 2004 to 2016. Coefficients obtained using MODIS DOD and observed controlling
factors and those using CMIP5 multi-model mean DOD and controlling factors are
shown in the left and right columns, respectively. The color of the shading denotes the
largest coefficient in absolute value among the three, while the saturation of the color
shows the magnitude of the coefficient (from 0 to 0.02). Only regression coefficients
significant at the 90% confidence level (Bootstrap test) are shown. Missing values are
shaded in grey. To highlight coefficients near the source regions, a mask of LAI ≤ 0.5 is
applied.

Figure 7. Projected changes of DOD in the late half of the 21st century (under the RCP
8.5 scenario) from that in the historical level (1861-2005) by CMIP5 multi-model mean
for four seasons. The percentage change of global mean (over land) DOD ± one inter-





pamong the models reaches 71.4% (i.e., at least five out seven models have the same sign
as the multi-model mean) are dotted.

Figure 8. Projected difference of (a)-(d) precipitation (mm day-1), (e)-(h) bareness, and
(i)-(l) 10 m wind (m s-1) between the late half of the 21st century (2051-2100; RCP 8.5
scenario) and historical level (1861-2005) from multi-model mean of seven CMIP5
models. Areas with sign agreement among the models reaches 71.4% (i.e., at least five
out seven models have the same sign as the multi-model mean) are dotted.

Figure 9. Projected change of DOD in the late half of the 21st century under the RCP 8.5
scenario by the regression model. The results are calculated using the regression
coefficients obtained from observations during 2004-2016 (see methodology) and
projected changes of precipitation, bareness, and surface wind from 16 CMIP5 models.
Dotted areas are regions with sign agreement among the models above 62.5% (i.e., at
least 10 out 16 models have the same sign as the multi-model mean). To highlight DOD
variations near the source regions, a mask of LAI ≤ 0.5 (from present-day climatology) is
applied.

Figure 10. (a)-(d) Projected change of DOD in the late half of the 21st century under the
RCP 8.5 scenario by the regression model (same as Fig. 9), and contributions from each
component, (e)-(h) precipitation, (j)-(i) bareness, and (m)-(p) surface wind speed. Dotted
areas are regions with sign agreement among the models above 62.5%. To highlight



DOD variations near the source regions, a mask of LAI $\leq$ 0.5 (from present-day
climatology) is applied.























Table 1 CMIP5 models used in this study. Models tagged with plus signs (+) considered
anthropogenic land use/land cover change in their vegetation prediction.

| Model | lat/lon resolution | Dust emission scheme | Dynamic Vegetation | Model reference |
|---|---|---|---|---|
| CanESM2 | 2.8°×2.8° | Reader et al. (1999); Croft et al. (2005) | N[+] | Arora et al. (2011) |
| GFDL-CM3 | 2.0°×2.5° | Ginoux et al. (2001) | Y[+] | Donner et al. (2011) |
| HadGEM2-CC | 1.2°×1.8° | Marticorena and Bergametti (1995) | Y[+] | Collins et al. (2011) |
| HadGEM2-ES | 1.2°×1.8° | Marticorena and Bergametti (1995) | Y[+] | Collins et al. (2011) |
| MIROC-ESM | 2.8°×2.8° | Takemura et al. (2000) | Y[+] | Watanabe et al. (2011) |
| MIROC-ESM-CHEM | 2.8°×2.8° | Takemura et al. (2000) | Y[+] | Watanabe et al. (2011) |
| NorESM1-M | 1.9°×2.5° | Seland et al. (2008) | N[+] | Bentsen et al. (2013) |







Table 2 List of regions selected to compare model output with MODIS DOD. Locations
of these regions are also plotted in Fig. 1b. Acronyms are used for some regions for short,
and are listed in the brackets in the first column. Note that the region names such as
Northern China and India are not exactly the same as their geographical definitions but
also covers some areas from nearby countries.

| Region | Domain |
| --- | --- |
| North Africa (N. Africa) | 5°-50°N, 18°W-35°E |
| Middle East | 12°-50°N, 35°-60°E |
| Northern China (N. China) | 35°-50°N, 70°-110°E |
| North America (N. America) | 25°-50°N, 95°-125°W |
| India | 5°-35°N, 60°-90°E |
| Southeastern Asia (SE. Asia) | 9°-35°N, 90°-121°E |
| South Africa (S. Africa) | 15°-35°S, 10°-50°N |
| South America (S. America) | 0°-55°S, 60°-83°W |
| Australia | 10°-40°S, 112°-155°E |




Table 3 Changes of DOD in the late half of the 21[st] century (2051-2100; RCP 8.5
scenario) from the historical condition (1861-2005) projected by CMIP5 multi-model
mean (second to fifth columns) and the regression model (sixth to ninth columns) in nine
regions. Changes of DOD are shown in percentage with reference to CMIP5 multi-model
historical run. Note that in some regions the projected change by the regression model is
quite large (i.e., greater than ± 100%), largely due to the underestimation of CMIP5
historical run in these regions.

| Region | CMIP5 | | | | Regression model | | | |
|---|---|---|---|---|---|---|---|---|
| | DJF | MAM | JJA | SON | DJF | MAM | JJA | SON |
| N. Africa | -3.8 | -3.6 | 2.4 | -16.3 | -0.8 | -17.7 | 11.1 | -10.3 |
| Middle East | 7.8 | 4.5 | 6.4 | 1.5 | 9.8 | -16.0 | -5.4 | -8.4 |
| N. China | -33.5 | -11.4 | -9.8 | -14.4 | 312.3 | -238.6 | -51.2 | -30.0 |
| N. America | 42.6 | 26.8 | 13.2 | -6.4 | -38.5 | -90.0 | 9.3 | -42.4 |
| India | -5.1 | 0.2 | -1.0 | -9.9 | -27.6 | -8.2 | -2.9 | -32.3 |
| SE. Asia | -45.7 | -16.5 | -13.5 | -17.1 | -34.8 | 1.6 | 4.2 | 96.3 |
| S. Africa | 24.0 | 6.1 | 38.5 | 54.4 | 22.3 | 59.3 | 231.8 | 78.3 |
| S. America | 35.7 | 27.4 | 51.8 | 36.0 | 14.8 | 56.1 | 78.3 | 154.6 |
| Australia | -3.2 | -3.2 | 15.3 | 17.0 | 2.7 | 0.4 | 0.7 | 3.7 |






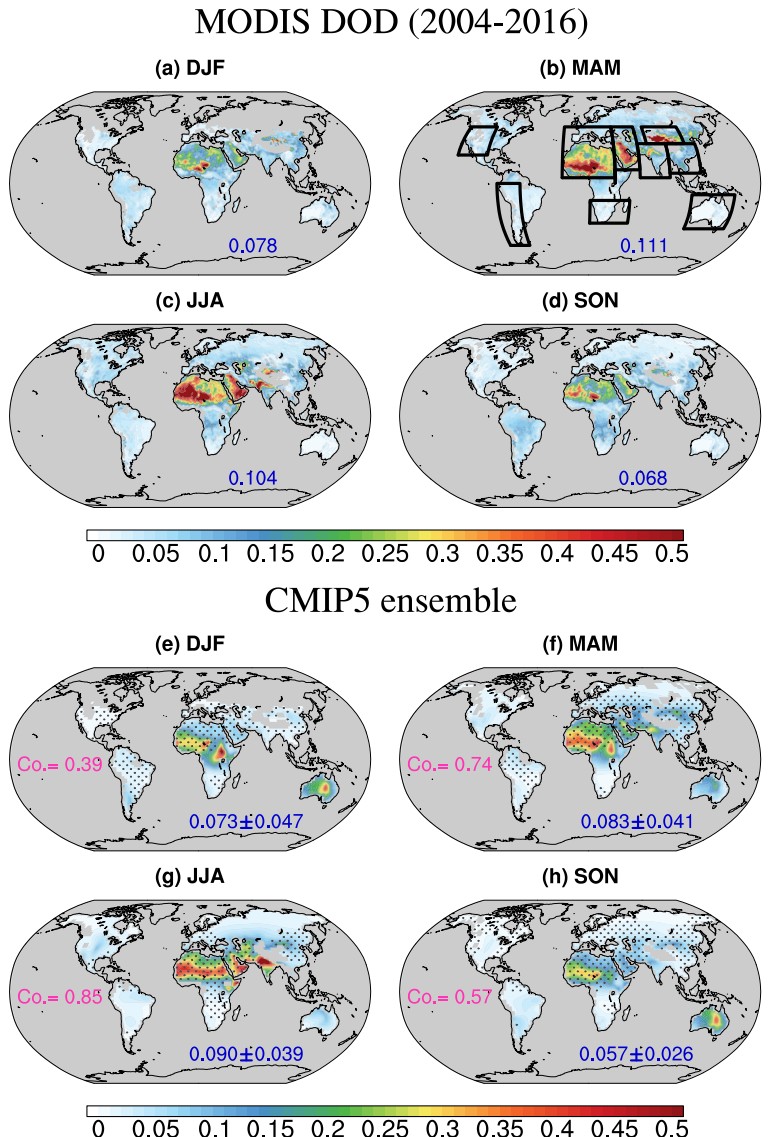

Figure 1. Climatology (2004-2016) of Aqua and Terra combined DOD (i.e., MODIS DOD; top panel) and multi-model mean of CMIP5 DOD (bottom) for four seasons. The pattern correlation (centered; calculated after interpolating MOIDS DOD to CMIP5 DOD grids) between CMIP5 and MODIS DOD are shown in pink in the bottom panel. Blue numbers denote global mean DOD over land. For CMIP5 model results, ± one standard deviation among seven CMIP5 models is also shown. Black boxes in (b) denote nine averaging regions (Table 2). Here we only added these boxes in (b) instead of every plot to keep the figure clean. Note that CMIP5 multi-model mean is masked by MODIS DOD for comparison. Dotted area in (e)-(h) shows where multi-model mean is greater than one inter-model standard deviation.





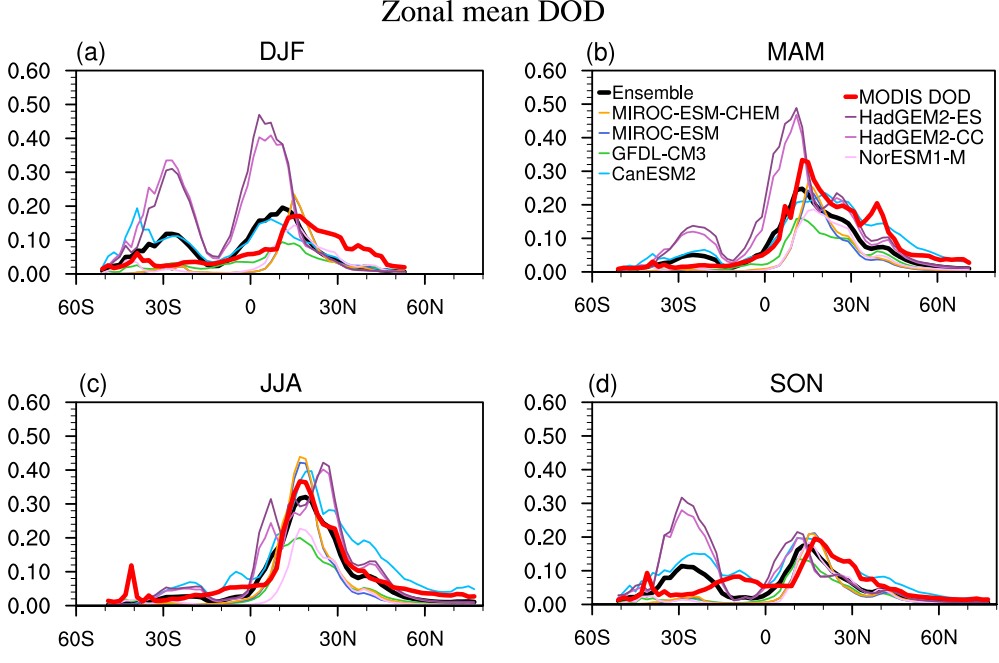

Figure 2. Zonal mean DOD from MODIS (thick red), CMIP5 multi-model mean (thick black), and each individual model (other colorful lines).



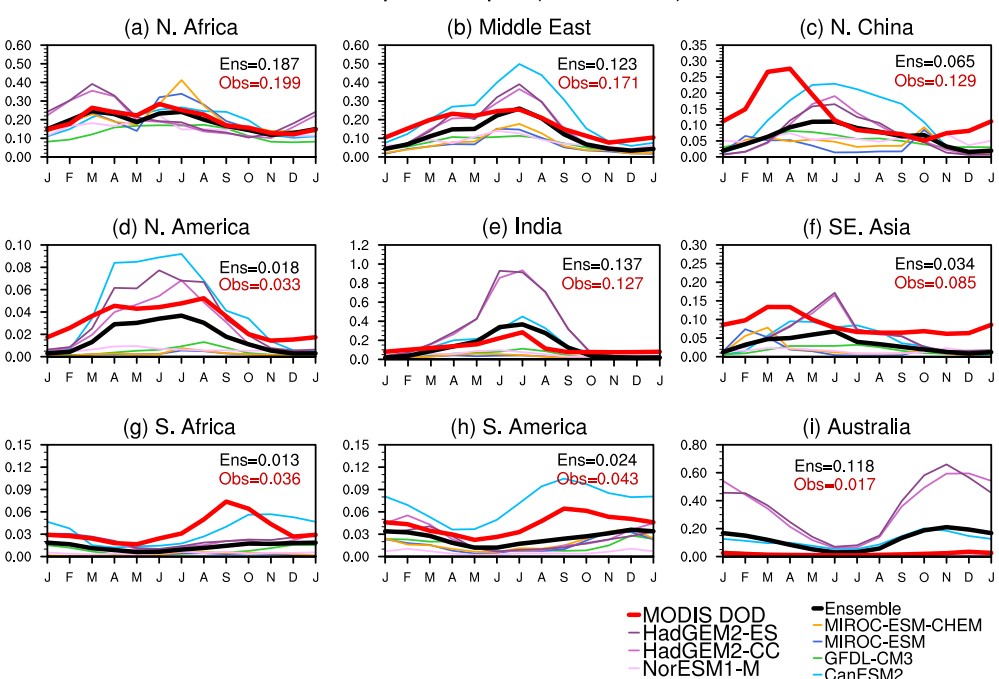

Figure 3. Seasonal cycle of DOD in nine regions (Table 2) averaged over 2004-2016.
Thick red lines denote MODIS DOD, thick black lines denote CMIP5 multi-model mean,
and other colorful lines denote individual model output. The annual means from MODIS
DOD (Obs; red) and multi-model mean (Ens; black) are also listed in each panel.





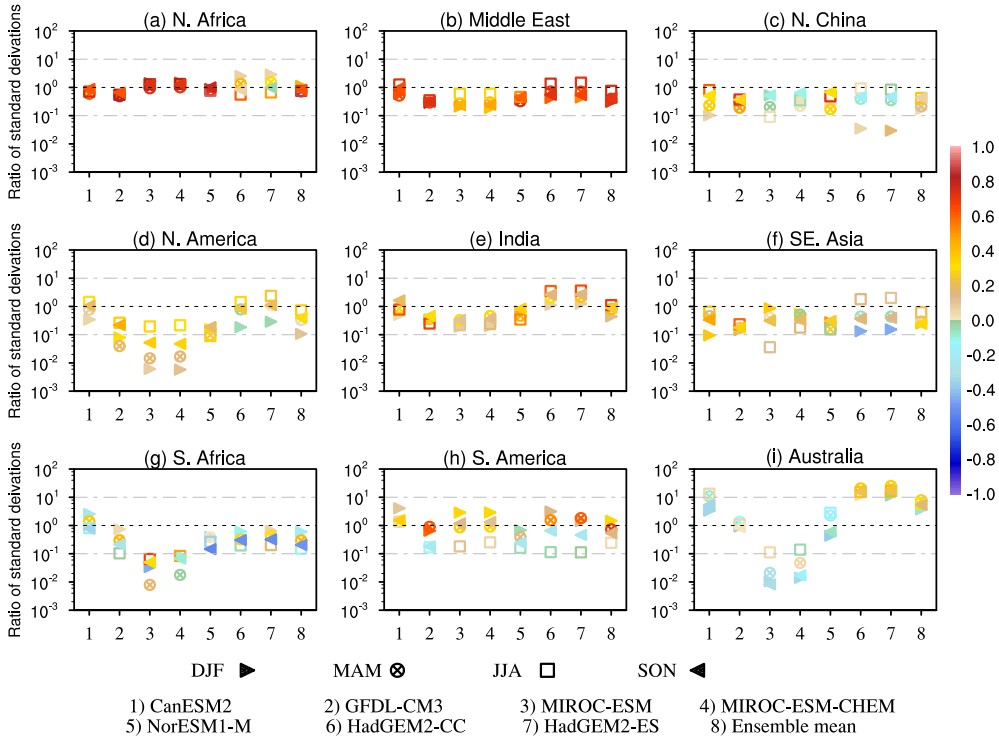

Figure 4. Spatial statistics comparing DOD from CMIP5 models with that from MODIS in nine regions. Label on the X-axis shows individual models (1-7) and multi-model mean (8). Y-axis shows the ratio of pattern standard deviations between model climatology (2004-2016) and that of MODIS, which reveals the relative amplitude of the simulated DOD versus satellite DOD. The color denotes pattern correlation (centered) between each model and MODIS DOD in each region.





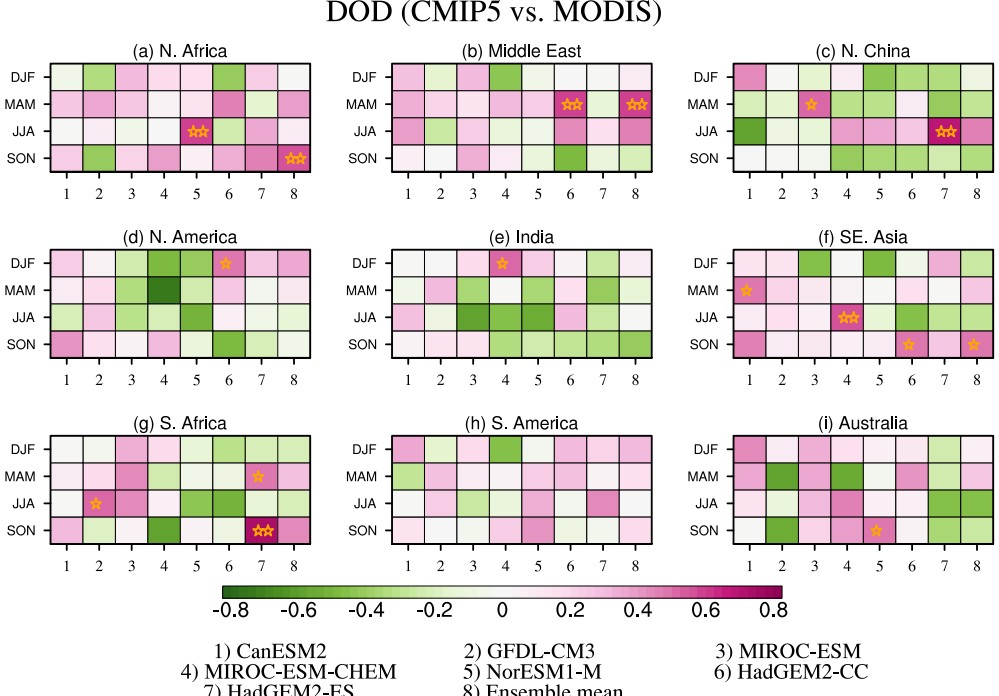

Figure 5. Correlations (color) between regional averaged time series from CMIP5 DOD and MODIS DOD from 2004 to 2016 for four seasons. Numbers in the X-axis denotes each model (1-7) and multi-model mean (8). Correlations significant at the 90% confidence level are marked by a star and significance at the 95% confidence level by two stars.



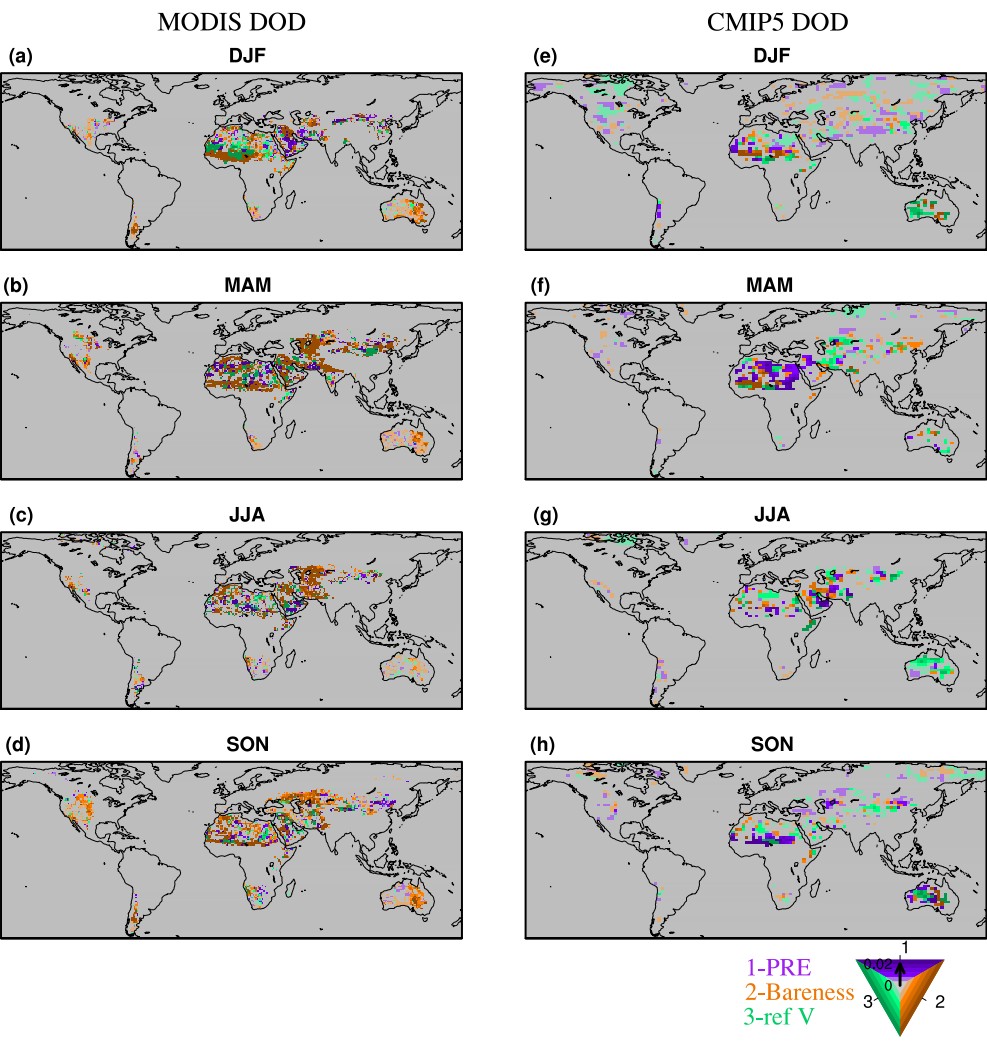

Figure 6. Regression coefficients calculated by regressing DOD in each season onto
standardized precipitation (purple), bareness (orange), and surface wind speed (green)
from 2004 to 2016. Coefficients obtained using MODIS DOD and observed controlling
factors and those using CMIP5 multi-model mean DOD and controlling factors are
shown in the left and right columns, respectively. The color of the shading denotes the
largest coefficient in absolute value among the three, while the saturation of the color
shows the magnitude of the coefficient (from 0 to 0.02). Only regression coefficients
significant at the 90% confidence level (Bootstrap test) are shown. Missing values are
shaded in grey. To highlight coefficients near dust source regions, a mask of LAI ≤ 0.5 is
applied.





Changes of CMIP5 DOD (2051-2100 minus 1861-2005)

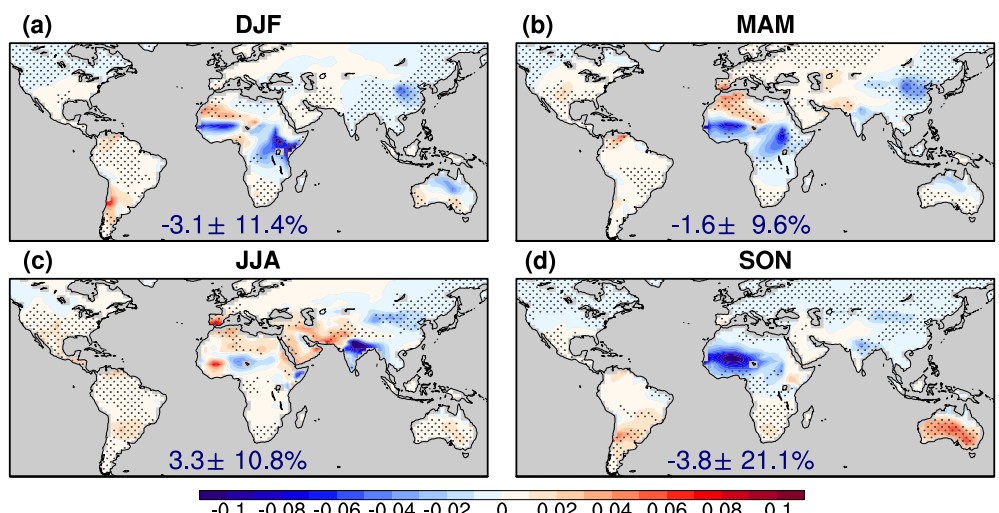


Figure 7. Projected changes of DOD in the late half of the 21$^{st}$ century (under the RCP
8.5 scenario) from that in the historical level (1861-2005) by CMIP5 multi-model mean
for four seasons. The percentage change of global mean (over land) DOD ± one inter-
model standard deviation is shown at the bottom of each plot. Areas with sign agreement
among the models reaches 71.4% (i.e., at least five out seven models have the same sign
as the multi-model mean) are dotted.


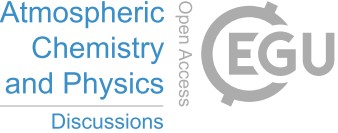

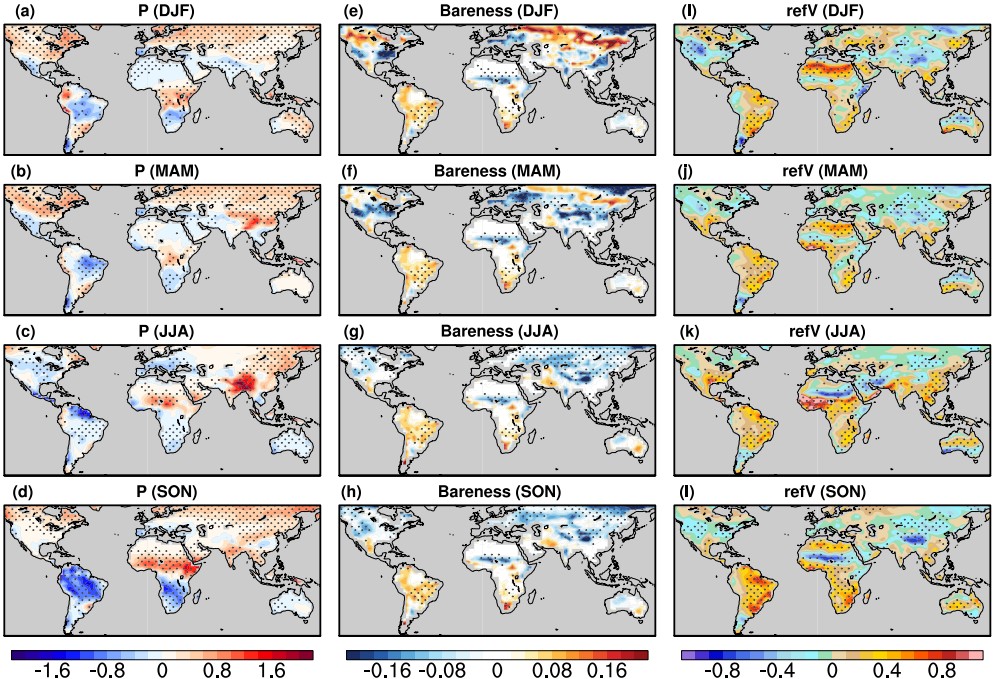


Figure 8. Projected difference of (a)-(d) precipitation (mm day$^{-1}$), (e)-(h) bareness, and
(i)-(l) 10 m wind (m s$^{-1}$) between the late half of the 21$^{st}$ century (2051-2100; RCP 8.5
scenario) and historical level (1861-2005) from multi-model mean of seven CMIP5
models. Areas with sign agreement among the models reaches 71.4% (i.e., at least five
out seven models have the same sign as the multi-model mean) are dotted.






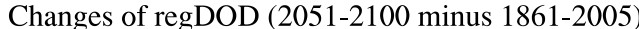

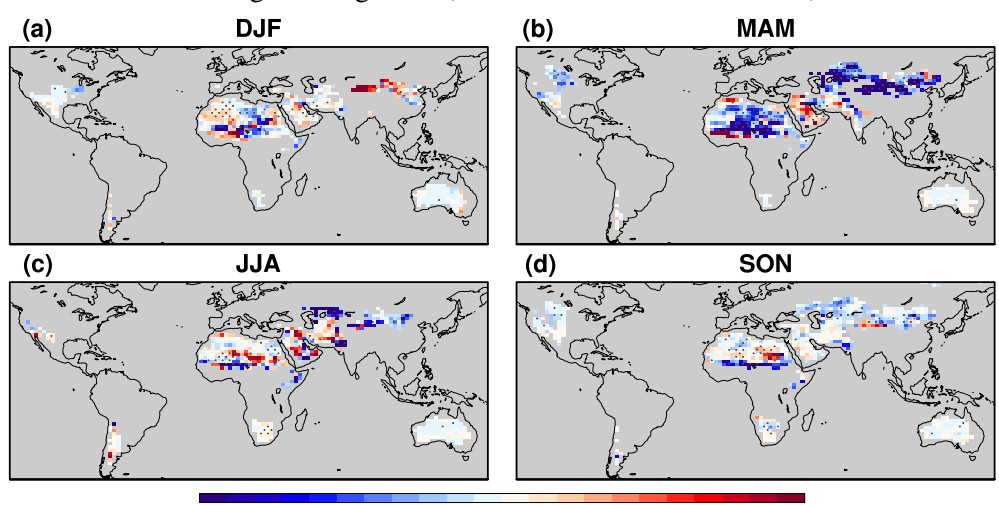

Figure 9. Projected change of DOD in the late half of the 21$^{st}$ century under the RCP 8.5
scenario by the regression model. The results are calculated using the regression
coefficients obtained from observations during 2004-2016 (see methodology) and
projected changes of precipitation, bareness, and surface wind from 16 CMIP5 models.
Dotted areas are regions with sign agreement among the models above 62.5% (i.e., at
least 10 out 16 models have the same sign as the multi-model mean). To highlight DOD
variations near the source regions, a mask of LAI ≤ 0.5 (from present-day climatology) is
applied.





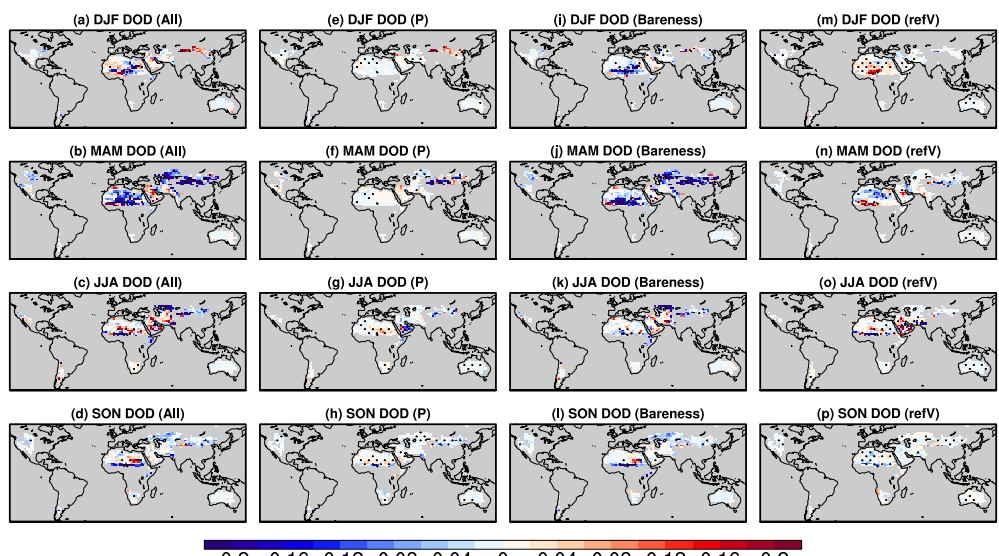

Figure 10. (a)-(d) Projected change of DOD in the late half of the 21st century under the
RCP 8.5 scenario by the regression model (same as Fig. 9), and contributions from each
component, (e)-(h) precipitation, (j)-(i) bareness, and (m)-(p) surface wind speed. Dotted
areas are regions with sign agreement among the models above 62.5%. To highlight
DOD variations near the source regions, a mask of LAI $\leq$ 0.5 (from present-day
climatology) is applied.