# Peer review of "How reliable are CMIP5 models in simulating dust optical depth?"

_Atmospheric Chemistry and Physics, 2018_

## Referee Comment (RC1) · Anonymous Referee #1 · 10 May 2018

This work examines the performance of seven CMIP5 climate models with interactive dust emissions schemes against dust optical depth (DOD) from MODIS Deep Blue aerosol products. The performance assessment to reproduce magnitude, spatial pattern and variations of observed DOD is conducted in nine regions, namely North Africa, Middle East, Northern China, North America, India, Southeastern Asia, South Africa, South America and Australia. Furthermore, interannual variations of DOD are also examined together with the impact on it of controlling factors such as 10 m surface wind, precipitation and surface bareness derived from leaf are index (LAI) data. In order to examine the relative contribution of these controlling factors to DOD multiple linear regression is applied on both, observations and models. Calculated regression coefficients in addition to observed and simulated controlling factors are then used to project

[Figure]

DOD to the future (both observations and models).

The authors show that although the models can reproduce the global distribution of DOD over land under present conditions, with a better representation over northern than southern hemisphere, the interannual variability of DOD is all in all not well captured by CMIP5 models. Furthermore, models also do not reproduce the observed relations between the DOD and the examined controlling factors. Projected changes of CMIP5 model mean under the RCP8.5 scenario are presented and compared to projections of a regression model.

The research presented is interesting and the paper is well written. As the authors mention in their introduction, performance of CMIP5 models to simulate dust has received little attention and this work is a good first step to change this. I recommend this paper to be published in ACP after some comments have been addressed.

General Comments:

1. The authors highlight the importance of examining the performance of current climate models in simulating dust and they choose to assess this performance by evaluating simulated DOD. In fact, in lines 73-75 the authors claim that evaluating DOD in "CMIP5 models will provide a clear picture of model capability of dust simulation". Although optical depth is a very common variable when it comes to validate models with respect to aerosols (be it dust or any other species), it is an integrated variable and therefore it does not provide any insight into the performance to reproduce the vertical distribution of aerosols. It has been shown that regional and global dust models can present similar performance in simulating AOD but present large diversity in emissions, deposition, surface concentration and vertical distribution (Huneeus et al., 2016). Although that study refers to forecast application, it is consistent with the findings in Huneeus et al. (2011). The authors should acknowledge this limitation in the discussion or conclusions, that although this evaluation is informative and necessary, it does not provide a full picture of current climate models to simulate dust. This similar

performance in optical depth compared to large diversity in other parameters such as emissions, deposition and surface concentration might be linked to the practice to use AOD to tune dust simulations. Is this a practice that is also used in climate models?

2. In addition to examining the DOD projections from CMIP5 models, the authors also project DOD using calculated regression coefficients and compare these results to the simulated ones. I have to admit that I have difficulties in seeing the usefulness of this exercise. What is the point of it? The authors state that similarities are found between both projections "which may be informative" without specifying for what they might me informative. What do differences and similarities of both projections tell us?

3. The authors could improve the description of the methodology applied in the study. Regression coefficients are computed by regressing DOD from MODIS onto the observed controlling factors, the same procedure is repeated with model outputs to obtain "model" regression factors. Now when the interannual variability is examined, in line 320 it is unclear whether the reconstructed DOD using model regression factors or the one based on observations. I would have thought the former but then lines 332-335 refer to the observations making me doubt what reconstruction is then used in the analysis. Furthermore, regression analysis on observations is done at 1°x1° resolution (lines 207-208) while for model outputs the regression analysis is done at 2°x2.5° resolution. But at what resolution are the reconstructed projections done? at the observation or the model resolution? Potential impacts on the regression coefficients due to different resolution should also briefly be discussed.

4. I find it confusing that the paper is build around the seven CMIP5 models with interactive dust emissions to examine their performance to simulate DOD. But when presenting and describing the projections, the reconstructed ones based on the 16 models are considered. I understand and agree with the authors in the reasons to include more models, but then I would have expected that when examining the model performance (both climatology and interannual variability) these reconstruction (from the 16 models) also would be considered in order to be able to draw any conclusion

from their projections. How good do these reconstructed projections (16 models) perform when compared with observations in present conditions? Sure, outputs of figure 9 and S8 are similar, but are they for the same reasons? Unfortunately analysis in figure 6 cannot be reproduced for the 16 models. Maybe it would make more sense to base results with respect to reconstructed projections in section 3.3 on figure S8 and move current figure 9 to the supplement (basically swaooing as it is now) and then build on how these results are also seen (or not) in the 16 models.

Specific Comments:

Page 5, lines 73-75: See general comment above, I would suggest reformulating the statement.

Page 6, lines 98-100: Given the importance of DOD in this study I suggest you briefly describe the method how DOD was derived from AOD and specify the modifications you applied to adapt the method to collection 6.

Page 7, line 134: Table 2 is referenced without any reference to Table 1. At present Table 1 corresponds to information on the models used in this study which is addressed in section 2.3. Tables should be arranged according to the order they are referenced in the text.

Page 8, lines 138-143: Surface wind speed, bareness and precipitation are defined as controlling factors without providing any evidence or explanation why these parameters. However in lines 321-331 the authors explain why these parameters have been selected. I suggest moving these lines forward to section 2.2.

Page 8, line 156: Remove PRECL.

Page 9, lines 164-182: A reference to (current) Table 1 should be made in this section. In addition, information on the 16 models used in the future projections needs to be provided.

Page 10, line 188: Provide a reference for the mass extinction efficiency used.

Page 10, lines 191-201: The authors illustrate the difference between the derived DOD and simulated one from one of the seven CMIP5 models with interactive dust emissions. It seems arbitrary why this model is used and not any other of the seven models? Is the intention of these lines to validate the derived DOD and therefore the chosen method? If that's the case then a more thorough validation should be done such as comparing the derived model mean DOD from all 16 models to the model mean from the seven CMIP5 models. Otherwise I don't see the point of having these analysis.

Page 11, lines 216-220: What period is considered in this analysis, same as observations, ie 2004-2016?

Page 11, line 226: Please provide some information on these 16 models, which models are they? Are the seven model with interactive dust emission part of these 16 models? Do they have prescribed emissions? A similar table as Table 1 should be included with relevant information of these 16 models.

Page 13, lines 258-260: How do the authors explain the shift to the north in the DOD by HadGEM2?

Page 13, line 269: remove "than other seasons".

Page 13, line 270: add "by the model mean" after "captured".

Page 13, lines 269-271: Since individual models are illustrated, authors should not only focus on the multi model mean but also on the individual models and their differences with respect to the multi model mean and the observations. For instance, MIROC and GFDL do not present the observed variability, in particular over N. America and India and they also present a different variability than the other models over northern China, with the peak closer to the observed one.

Page 14, lines 276-277: The MODIS DOD peak in Australia is hardly seen.

Page 15-16, lines 319-321: Which reconstructed DOD is used here? is it the one considering observed regression coefficients and simulated controlling factors? Or is

it the one using simulated regression coefficients derived from model DOD and model controlling factors? Also, are only the seven CMIP5 models with interactive dust emission considered? The authors should be more specific which reconstruction they refer. Also, couldn't the correlation based on reconstructed DOD be integrated in the figure as an additional column?

Page 16, lines 321-332: move these lines to section 2. See general comment.

Page 18, lines 378-380: I have difficulties seeing the similarities in North Africa and the Middle east between the MODIS and CMIP5 regression coefficients pointed out by the authors. I actually see more the differences between both regressions in both regions. I would suggest the authors review the analysis in these lines.

Page 22, lines 470-473: On which results are the authors basing this statement. I suggest specifying.

Pages 22-23, lines 465-490: These lines would fit better in the discussion section.

Page 24, line 522-524: The statement seems something that would fit better in the conclusion section. Consider moving it.

Page 25, line 546: Suggest replacing "quite well" with something more academic.

Page 27, line 583: In which way are similarities between both projections "informative"? What information do they provide.
* * *

---

## Referee Comment (RC2) · Anonymous Referee #2 · 12 May 2018

The article presents an in-depth analysis of the CMIP5 models' ability to reproduce the dust optical depth (DOD), considering both seasonal and inter-annual variability, as well as the driving factors behind those DOD levels. The observational data used are DOD over land derived from MODIS Terra-aqua data; bareness derived from AVHRR; 10m wind speed from ERA-Interim reanalysis; and precipitation from PRECL.

The analysis of the driving factors is performed by regressing the observed DOD from MODIS over land to the observed/reanalyzed driving factors. The analysis is then extended to future climate scenarios (RCP8.5) using both the CMIP5 models' dust outputs and the regression based on present day observed relationships between DOD and the driving factors.

The main results/conclusions are: 1) Models behave better over the NH large dust

sources. 2) Models do not reproduce interannual variability 3) The constraints from bareness in models are underestimated and the influences of wind speed and precipitation are overestimated. 4) A corrected projection of DOD based on the regression model is proposed. There are some similarities between the projections and the corrected projections.

The paper is very interesting, includes novelties and deserves publication. However, I have several doubts and comments that need clarification and further discussion.

General comments

1) DOD from MODIS: It is not clear what the DOD derived from MODIS refers to. Is it total dust optical depth or coarse dust optical depth? I understand that it refers to the total dust optical depth (fine and coarse) but I was confused when the product was compared to the coarse (O'Neill) product from AERONET. Can you please explain better the derivation of DOD from AOD in the paper? Given the importance of the dataset for the paper I feel it is not enough to refer the reader to other publications. Also, can you provide an estimation of the uncertainty of this product? The confidence of satellite data over the different regions is assessed by comparison with AERONET (few stations, low spatial coverage), CALIOP, and considering the number of days with available DOD per season. Results show that while in Africa, South America, Middle East and some Asian regions confidence seems to be high, for some regions in Asia/North America it largely depends on the season. Âă In my view, the strength or confidence on the DOD data by region should be considered when discussing: the modelled DOD evaluation at the regional level, the regression method projections and discrepancies with CMIP5 models. Âă

2) DOD from CMIP5 models: The authors compare the DOD derived from the selected CMIP5 models using Eq. (2). Using a value of 0.6 everywhere and for every model is an important simplification as it depends on model-dependent assumptions on size distribution and other issues such as the size range considered. While 0.6 may be

a reasonable value for GFDL-CM3, how can we be sure it is ok for other models? Is there any other model for which you could compare this assumption in addition to GFDL-CM3.

3) Clear sky vs all sky values: While the authors have made an effort to gather the largest possible amount of DOD data by using both Aqua and Terra, the results of the comparison between MODIS DOD and model DOD may be quite affected by the use of all sky values from the models instead of clear sky values. Can you at least quantify this effect by for example using clear sky DOD from GFDL-CM3? How large is this effect? This may be potentially important in areas with seasonal clouds and precipitation. Could this be one of the reasons for the strong disagreement in some regions?

4) Interannual variability: One of the findings of this study is that the interannual DOD variation is not very well captured by the CMIP5 models. It is stated that "models probably misrepresented these [controlling factor] relationships, in addition to their incapacity of capturing the interannual variations of individual controlling factors". Because of their nature, CMIP5 models cannot (and are not meant to) represent year-to-year variations of the driving factors in such a short time period. Therefore, the first part of the statement is just speculative, i.e., one cannot know whether the relationships are misrepresented from that analysis alone. I strongly believe that this part should be better discussed both in the results section and the conclusions. I also believe that the comparison between CMIP5 model output and observations in Figures S4 to S6 is not needed. Isn't it obvious that CMIP5 models cannot represent year-to-year variations of each season in a 12-year period?

5) The role of surface bareness: one of the important conclusions of the study is that "constraints from surface bareness are largely underestimated while the influences of surface wind and precipitations are overestimated". I have a few doubts/comments on this:

a. How can you know that the constraint from surface bareness is largely underestimated? Given your methodology, couldn't it be that the constraint of surface bareness is correct in absolute terms but the effect of precipitation (through soil humidity) is overestimated? This should be clarified. b. While I think that the methodology is sound, it is not clear to me how year-to-year variations of around 2-3 % in LAI (Figure S7) can have such an impact in the interannual variability of dust in Northern Africa. Because this conclusion has important implications, could you further discuss this point? What would be the physical mechanisms that could explain this? Can you provide the same figure (S7) but for the model derived bareness (both present day and future projections)? How well do the models compare with the observed range of variability of the LAI in arid regions (the Sahara for example)?

6) Regression method projections vs. CMIP5 projections: The regression method used to derive DOD in future scenarios is based upon 16 CMIP5 model variables (surface wind speed, bareness and precipitation) and compared to dynamical projections of only 7 CMIP5 models (those with online dust schemes). Partly, differences in future trends might come by differences in driving variables. You state [line 439] that projected DOD changes using the full sample or only 7 models are very similar. If so, why not using the same 7 model outputs as drivers? This would enhance consistency. Finally, why the similarities between the two approaches in some regions may be informative?

Minor comments:

- I suggest to list multiple references to the same topic chronologically, unless there are reason to order them differently, e.g. in the introduction.

- I think the column heading "Dust emission scheme" is somewhat misleading as the given references describe the implementation of a dust emission scheme, not the scheme itself. Perhaps rewording to "Dust emission implementation" or similar would help.

- I suggest changing Eq. (1) to Bareness = exp(-LAI). Also, is there a reference for this

equation?

- L. 146-147: I would normally not consider a resolution of 80km "very suitable to study the influence of wind speed on dust emission and transport on small scales". I understand the intent of this statement, but I suggest rephrasing this to avoid misunderstanding.

- L. 160: I suggest to delete "relatively high" as well as "quite"

- L. 192: GFLD-CM3 should be GFDL-CM3Âă

- Line 205: clarify which DOD is regressed onto observed values, i.e. satellite derived DODÂă

- Fig. 3i: It is very hard to see the MODIS DOD pattern for Australia. Can this be improved?

- Fig. 4 is ok, but quite dense

- L. 311: variability instead of variations

- L. 328: wind erosion instead of "soil erosion from wind"

- Line 431: centaury should be centuryÂă

- L. 457 ff: Sometimes it is not clear if "models" refers to the CMIP5 models or projection 'models'.

- Figure 6. It is difficult to sort out the different elements, e.g. the strength of the regression depending on the shading intensity is not visible. I would suggest: to make a zoom per region, or to display dependencies from the 3 variables in independent maps, and to use the same resolution for MODIS and CMIP5 maps to make easier a direct comparison.Âă

---

## Author Comment (AC1) · 14 Jun 2018

We thank the reviewer for very helpful comments. We reply to your comment (in Italic) below.

*This work examines the performance of seven CMIP5 climate models with interactive dust emissions schemes against dust optical depth (DOD) from MODIS Deep Blue aerosol products. The performance assessment to reproduce magnitude, spatial pattern and variations of observed DOD is conducted in nine regions, namely North Africa, Middle East, Northern China, North America, India, Southeastern Asia, South Africa, South America and Australia. Furthermore, interannual variations of DOD are also examined together with the impact on it of controlling factors such as 10 m surface wind, precipitation and surface bareness derived from leaf are index (LAI) data. In order to examine the relative contribution of these controlling factors to DOD multiple linear regression is applied on both, observations and models. Calculated regression coefficients in addition to observed and simulated controlling factors are then used to project DOD to the future (both observations and models).*
*The authors show that although the models can reproduce the global distribution of DOD over land under present conditions, with a better representation over northern than southern hemisphere, the interannual variability of DOD is all in all not well captured by CMIP5 models. Furthermore, models also do not reproduce the observed relations between the DOD and the examined controlling factors. Projected changes of CMIP5 model mean under the RCP8.5 scenario are presented and compared to projections of a regression model.*
*The research presented is interesting and the paper is well written. As the authors mention in their introduction, performance of CMIP5 models to simulate dust has received little attention and this work is a good first step to change this. I recommend this paper to be published in ACP after some comments have been addressed.*

*General Comments:*
*1. The authors highlight the importance of examining the performance of current climate models in simulating dust and they choose to assess this performance by evaluating simulated DOD. In fact, in lines 73-75 the authors claim that evaluating DOD in "CMIP5 models will provide a clear picture of model capability of dust simulation". Although optical depth is a very common variable when it comes to validate models with respect to aerosols (be it dust or any other species), it is an integrated variable and therefore it does not provide any insight into the performance to reproduce the vertical distribution of aerosols. It has been shown that regional and global dust models can present similar performance in simulating AOD but present large diversity in emissions, deposition, surface concentration and vertical distribution (Huneeus et al., 2016). Although that study refers to forecast application, it is consistent with the findings in Huneeus et al. (2011). The authors should acknowledge this limitation in the discussion or conclusions, that although this evaluation is informative and necessary, it does not provide a full picture of current climate models to simulate dust. This similar*

*performance in optical depth compared to large diversity in other parameters such as emissions, deposition and surface concentration might be linked to the practice to use AOD to tune dust simulations. Is this a practice that is also used in climate models?*

We thank the reviewer for pointing out that DOD cannot provide a full picture of dust modeling skill by CMIP5 models. We modified lines 75-77: "A comprehensive evaluation of the climatology and interannual variation of global dust optical depth (DOD) in CMIP5 models will provide insights into models' capability of simulating the integrated aerosol extinction due to dust, which is one of the key variables that determine radative forcing of dust to the climate system." and lines 604-609: "Since DOD is an integrated variable, it does not reflect the vertical distribution of dust aerosols. As pointed by Huneeus et al., (2016), dust models with similar performance in simulating aerosol optical depth may have quite large differences in simulating vertical distribution, emission, deposition, and surface concentration of dust. An overall evaluation of dust modeling capability will require detailed examination of these variables and the life cycle of dust in CMIP5 models in addition to DOD." to better address this issue.

We agree with the reviewer that the similar performance of models in simulating DOD versus their discrepancies in simulating variables such as surface concentration, emission, and deposition may be due to the fact that DOD or AOD is used to tune dust models. Same tuning method may be used in the climate models, too, and thus adds to the need to examine other variables related to dust life cycle in the CMIP5 models.

*2. In addition to examining the DOD projections from CMIP5 models, the authors also project DOD using calculated regression coefficients and compare these results to the simulated ones. I have to admit that I have difficulties in seeing the usefulness of this exercise. What is the point of it?*

The reason to provide a future DOD projection by the regression model in addition to CMIP5 models' projection was not clearly addressed in the previous version. We added lines 513-522 to better explain the purpose of this analysis: "Here we also present the projected change of DOD from the regression model in Figure 9. The regression model (see section 2.4 for details) is developed based on observed relationships between MODIS DOD and local controlling factors and can largely capture the interannual variations of DOD in the present-day climate (Table S1 in the Supplement). Assuming that the observed connection between DOD and these controlling factors do not change dramatically in the future, we can use this regression model and CMIP5-model projected change of controlling factors to project DOD variations. Compared to DOD projection from CMIP5 models, this approach utilizes additionally observational constrains and is likely to provide a more reliable future projection."

*The authors state that similarities are found between both projections "which may be informative" without specifying for what they might me informative. What do differences and similarities of both projections tell us?*

We removed "which may be informative", and modified the sentence to: "we find some similarities between the two, which adds to the confidence of projected DOD change in these regions..." (lines 690-691). Although CMIP5 models overestimate the influence of surface wind and precipitation and underestimate the role of bareness, there are some similarities between model and observations over regions such as North Africa in DJF and parts of the Arabian Peninsula in JJA (Fig. 6; lines 450-462), which indicate that models partially capture the connection between the DOD and these controlling factors in some regions. Therefore, the projection of DOD from CMIP5 models (Fig. 7) is not completely unreliable. The similarity between CMIP5 projection and the projection from the regression model thus adds to the confidence of projected change of DOD over North Africa, the Arabian Peninsula, and northern China in some seasons.

*3. The authors could improve the description of the methodology applied in the study. Regression coefficients are computed by regressing DOD from MODIS onto the observed controlling factors, the same procedure is repeated with model outputs to obtain "model" regression factors. Now when the interannual variability is examined, in line 320 it is unclear whether the reconstructed DOD using model regression factors or the one based on observations. I would have thought the former but then lines 332-335 refer to the observations making me doubt what reconstruction is then used in the analysis.*

The regression coefficients are derived from observations. We modified section 2.4 in the methodology section and lines 412-414 to improve the clarity.

*Furthermore, regression analysis on observations is done at 1_x1_ resolution (lines 207-208) while for model outputs the regression analysis is done at 2_x2.5_ resolution. But at what resolution are the reconstructed projections done? at the observation or the model resolution? Potential impacts on the regression coefficients due to different resolution should also briefly be discussed.*

For future projection, the regression coefficient is interpolated to a 2° by 2.5° grid to be consistent with model output. So the projected DOD is also on a 2° by 2.5° grid. We modified lines 282-284 to clarify this and discuss potential impacts of the interpolation: "The regression coefficients are interpolated from the 1° by 1° grid to a 2° by 2.5° grid to be consistent with model output. Such an interpolation may smooth out some spatial characteristics from observations."

*4. I find it confusing that the paper is build around the seven CMIP5 models with interactive dust emissions to examine their performance to simulate DOD. But when presenting and describing the projections, the reconstructed ones based on the 16 models are considered. I understand and agree with the authors in the reasons to include more models, but then I would have expected that when examining the model performance (both climatology and interannual variability) these reconstruction (from the 16 models) also would be considered in order to be able to draw any conclusion from their projections. How good do these reconstructed projections (16 models) perform when compared with observations in present conditions? Sure, outputs of figure 9 and S8 are similar, but are they for the same reasons? Unfortunately analysis in figure 6 cannot be reproduced for the 16 models. Maybe it would make more sense to base results with respect to reconstructed projections in section 3.3 on figure S8 and move current figure 9 to the supplement (basically swaooing as it is now) and then build on how these results*

*are also seen (or not) in the 16 models.*

We use seven CMIP5 model with interactive dust emission scheme because we would like to examine the relationship between DOD variations and local controlling factors, while in models with offline dust these connections are lost. We added lines 210-212 to better explain this. The purpose of using variables from 16 CMIP5 models for the future projection is to include as much information (i.e., more model output) about projected change of the controlling factors as possible.

We agree with the reviewer that it is better to show the future projection by the regression model and output from seven CMIP5 models in Figure 9 first and then discuss results from 16-model output later in Figure S7 in the Supplement. We followed the advice to switch the figures and modified text accordingly (lines 522-527, 549-557).

Here we also examine the climatology and interannual variations of reconstructed DOD (using 7-model output). The following figure shows the pattern correlation between MODIS DOD and reconstructed DOD using 7-model output and regression coefficients from observations. Figure R1a shows the pattern correlations between the climatologist of reconstructed DOD (regDOD) and MODIS DOD for 2004-2016 over 9 regions. The pattern correlations are very high, because the constant value in the regression model (i.e., $d$ in the equation $regDOD = a \times Precipitation + b \times Wind + c \times Bareness + d$) contains information from MODIS DOD, i.e., has a pattern similar to observed climatology.

We also show the anomalies of the reconstructed DOD where the influence of the constant value is largely removed. Figs. R1b-c show pattern correlations between MODIS DOD and regDOD for the differences of DOD between 2010-2016 and 1861-2005 (Fig. R1b) and between 2010-2016 and 2004-2016 (Fig. R1c). The latter (Fig. R1c) shows slightly better pattern correlations than the former (less green boxes) since the historical condition (1861-2005) is not exactly comparable with the 2004-2016 climatology. Fig. R1d shows the pattern correlation of MODIS and regDOD for the differences of DOD between 2010-2016 and 2004-2009. The pattern correlations are similar to Fig. R1c because relatively short time periods are used (7 years for the 2010-2016 mean and 6 years for the 2004-2009 mean) and values can be largely influenced by interannual variations of the controlling factors in the CMIP5 models.

[Figure]

Figure R1. Pattern correlations between MODIS DOD and reconstructed DOD (regDOD) that used output from seven CMIP5 models and observed regression coefficients for (a) 2004-2016 DOD climatology, the differences of DOD (b) between 2010-2016 and historical run, (c) between 2010-2016 and 2004-2016, (d) between 2010-2016 and 2004-2009 over nine regions. MODIS DOD anomaly during 2010-2016 (with reference to the 2004-2016 climatology) is used in calculating pattern correlations in both (b) and (c).

[Figure]

Figure R2. Correlations of regional averaged time series over nine regions between MODIS DOD and reconstructed DOD that used output from seven CMIP5 models and observed regression coefficients. Correlations significant at the 90% confidence level are marked by a star and significance at the 95% confidence level by two stars.

The correlations of regional averaged time series (2004-2016) between MODIS DOD and reconstructed DOD that used 7-model output and regression coefficients from observations are shown in Figure R2. As we mentioned in the paper, CMIP5 models are not expected to capture the interannual variations of the controlling factors, so we would not expect that the reconstructed DOD using CMIP5 output to capture the interannual variations of DOD, either. However, the variations of DOD over Africa in MAM, the Middle East in SON, India in MAM, and Australia in SON are to some extent captured by the regression model (Fig. R2). When we use observed controlling factors to reconstruct DOD (section 2.4.2), interannual variations during the present day is largely captured (Table S1).

The outputs of old Figs. 9 (from 16 models) and S8 (from 7 models) are similar because the projected changes of precipitation, surface wind speed, and bareness from 16-model ensemble mean (Fig. R3) show some features similar to 7-model ensemble mean (Fig. 8). We clarified this in the updated text (lines 549-557) and also added Fig. R3 to the supplement.

[Figure]

Figure R3. Projected difference of (a)-(d) precipitation (mm day$^{-1}$), (e)-(h) bareness, and (i)-(l) 10 m wind (m s$^{-1}$) between the late half of the 21$^{st}$ century (2051-2100; RCP 8.5 scenario) and historical level (1861-2005) from multi-model mean of 16 CMIP5 models. Areas with sign agreement among the models reaches 62.5% (i.e., at least ten out of 16 models have the same sign as the multi-model mean) are dotted.

*Specific Comments:*

*Page 5, lines 73-75: See general comment above, I would suggest reformulating the statement.*

   We modified those lines to: "A comprehensive evaluation of the climatology and interannual variation of global dust optical depth (DOD) in CMIP5 models will provide insights into models' capability of simulating the integrated aerosol extinction due to dust. DOD is also one of the key variables that determine radative forcing of dust to the climate system.". And also added discussion in lines 604-609 to acknowledge that DOD dose not reflect the vertical distribution of dust aerosols and more variables (such as surface dust concentration, emission, deposition, vertical distributions) are need to provide a whole picture of dust simulation in CMIP5 models.

*Page 6, lines 98-100: Given the importance of DOD in this study I suggest you briefly describe the method how DOD was derived from AOD and specify the modifications you applied to adapt the method to collection 6.*

   Lines 102-114 are added to describe how DOD is derived and adapted to collection 6.

*Page 7, line 134: Table 2 is referenced without any reference to Table 1. At present Table 1 corresponds to information on the models used in this study which is addressed in section 2.3. Tables should be arranged according to the order they are referenced in the text.*

   We actually referred Table 1 in line 78 when introducing the seven models used in this study.

*Page 8, lines 138-143: Surface wind speed, bareness and precipitation are defined as controlling factors without providing any evidence or explanation why these parameters. However in lines 321-331 the authors explain why these parameters have been selected. I suggest moving these lines forward to section 2.2.*

   We follow the advice to move lines 321-331 to section 2.2 (now lines 167-176).

*Page 8, line 156: Remove PRECL.*

   Here we refer to the precipitation data from PRECL and so will keep "PRECL precipitation".

*Page 9, lines 164-182: A reference to (current) Table 1 should be made in this section. In addition, information on the 16 models used in the future projections needs to be provided.*

   We added "Table 1" in line 207. We also modified lines 286-288 to clarify that information of 16 CMIP5 models can be found from the Supplementary Table S1 of Pu and Ginoux (2017).

*Page 10, line 188: Provide a reference for the mass extinction efficiency used.*

   The mass extinction efficiency used here is from Ginoux et al. (2012a) as mentioned in line 231. We also added discussion on this variable in lines 237-241.

*Page 10, lines 191-201: The authors illustrate the difference between the derived DOD and simulated one from one of the seven CMIP5 models with interactive dust emissions. It seems arbitrary why this model is used and not any other of the seven models? Is the intention of these lines to validate the derived DOD and therefore the chosen method? If that's the case then a more thorough validation should be done such as comparing the derived model mean DOD from all 16 models to the model mean from the seven CMIP5 models. Otherwise I don't see the point of having these analysis.*

In these lines we compare the derived DOD versus model calculated DOD in GFDL-CM3 to valid the method we used to derive DOD (i.e., Eq. e). We did not use this analysis to select models. We chose seven models with interactive dust emission schemes to examine DOD climatology and interannual variations because DOD in these models are influenced by environmental factors and the can be compared with observations, while in models with offline dust, these connections do not exist in the models.

We used GFDL-CM3 as an example to validate the DOD derivation because it's the only model among the seven that we can access model calculated DOD. We modified lines 241-253 to better present the analysis.

*Page 11, lines 216-220: What period is considered in this analysis, same as observations, ie 2004-2016?*

Yes. We modified line 271 to clarify this.

*Page 11, line 226: Please provide some information on these 16 models, which models are they? Are the seven model with interactive dust emission part of these 16 models? Do they have prescribed emissions? A similar table as Table 1 should be included with relevant information of these 16 models.*

Seven models are part of these 16 models. We modified line 286-288 to clarify that models information and dust emission schemes can be found from Supplementary Table S1 of Pu and Ginoux (2017).

*Page 13, lines 258-260: How do the authors explain the shift to the north in the DOD by HadGEM2?*

In lines 258-260 (original version) we referred the multi-model mean shown in Fig. 2b: "The peak around 19° N in North Africa and Middle East is well captured by the multi-model mean, although the magnitude is slightly underestimated." The overestimation of DOD around 28° N in the HadGeM2 model may be caused by its overestimation of DOD over the Middle East and India in summer (Figs. 3b, e).

*Page 13, line 269: remove "than other seasons".*

Done.

*Page 13, line 270: add "by the model mean" after "captured".*

Done.

*Page 13, lines 269-271: Since individual models are illustrated, authors should not only focus on the multi model mean but also on the individual models and their differences with respect to the multi model mean and the observations. For instance, MIROC and*

*GFDL do not present the observed variability, in particular over N. America and India and they also present a different variability than the other models over northern China, with the peak closer to the observed one.*

We revised lines 334-337, 345-347, 358-359 to add discussion on a few models' performance over North America, northern China, and Australia.

*Page 14, lines 276-277: The MODIS DOD peak in Australia is hardly seen.*

We have scaled MODIS DOD over Australia ten times in Fig. 3 and modified figure caption accordingly to better display the seasonal cycle of DOD.

*Page 15-16, lines 319-321: Which reconstructed DOD is used here? is it the one considering observed regression coefficients and simulated controlling factors? Or is it the one using simulated regression coefficients derived from model DOD and model controlling factors? Also, are only the seven CMIP5 models with interactive dust emission considered? The authors should be more specific which reconstruction they refer. Also, couldn't the correlation based on reconstructed DOD be integrated in the figure as an additional column?*

The reconstructed DOD used observed regression coefficients and observed controlling factors. We modified lines 398, 412-414 to clarify this. We actually considered adding a column to Fig. 5 to show the correlations between MODIS DOD and reconstructed DOD (see Fig. R4 below) in the early version of the paper. However, since the reconstructed DOD here used observed controlling factors, which make it slightly "unfair" to compare the results with those from CMIP5 DOD, we decide to present the results separately in Table 2.

[Figure]

Figure R4. Correlations (color) between regional averaged time series from CMIP5 DOD and MODIS DOD from 2004 to 2016 for four seasons. Numbers in the X-axis denotes each model (1-7), multi-model mean (8), reconstructed DOD (9). Correlations significant at the 90% confidence level are marked by a star and significance at the 95% confidence level by two stars.

*Page 16, lines 321-332: move these lines to section 2. See general comment.*
    Done.

*Page 18, lines 378-380: I have difficulties seeing the similarities in North Africa and the Middle east between the MODIS and CMIP5 regression coefficients pointed out by the authors. I actually see more the differences between both regressions in both regions. I would suggest the authors review the analysis in these lines.*
    Lines 458-462 are modified clarity this: "In JJA, the influences of precipitation and bareness over the eastern Arabian Peninsula in the multi-model mean (Fig. 6g) also show some similarity to observation (Fig. 6c), although an underestimation of the influence from bareness and an overestimation of precipitation are still there. "

*Page 22, lines 470-473: On which results are the authors basing this statement. I suggest specifying.*
    We added "in the present-day (Fig. 6)" after "Multi-model mean also overestimates the connection between DOD and precipitation and surface wind and underestimates the influence of bareness" to specify this argument. In section 3.2 we compared the multiple linear regression coefficients from CMIP5 models with those from the observations (Fig. 6) and found multi-model mean overestimates the connection between DOD and precipitation and surface wind while underestimates the influence of bareness.

*Pages 22-23, lines 465-490: These lines would fit better in the discussion section.*
    We prefer to discuss the uncertainties of CMIP5 and regression model projections right after showing the results of the two methods (Figs. 7-10) in section 3.3. In section 4, more general issues such as including other variables from CMIP5 models to examine model performance, studies on future dust projection, and the implication of the regression model, are discussed.

*Page 24, line 522-524: The statement seems something that would fit better in the conclusion section. Consider moving it.*
    This is not the key conclusion of the paper, so we prefer to keep it in the discussion.

*Page 25, line 546: Suggest replacing "quite well" with something more academic.*
    We modified the line to: "In JJA, the simulated zonal mean DOD from multi-model mean largely resembles MODIS DOD".

*Page 27, line 583: In which way are similarities between both projections "informative"? What information do they provide.*
    See our detailed reply to Comment #2. We removed "which may be informative", and modified the sentence to: "we find some similarities between the two, which adds to the confidence of projected DOD change in these regions, for instance..."
We thank the reviewer for very helpful comments. We reply to your comment (in Italic) below.

*The article presents an in-depth analysis of the CMIP5 models ability to reproduce the dust optical depth (DOD), considering both seasonal and inter-annual variability, as well as the driving factors behind those DOD levels. The observational data used are DOD over land derived from MODIS Terra-aqua data; bareness derived from AVHRR; 10m wind speed from ERA-Interim reanalysis; and precipitation from PRECL. The analysis of the driving factors is performed by regressing the observed DOD from MODIS over land to the observed/reanalyzed driving factors. The analysis is then extended to future climate scenarios (RCP8.5) using both the CMIP5 models' dust outputs and the regression based on present day observed relationships between DOD and the driving factors.*

*The main results/conclusions are: 1) Models behave better over the NH large dust sources. 2) Models do not reproduce interannual variability. 3) The constraints from bareness in models are underestimated and the influences of wind speed and precipitation are overestimated. 4) A corrected projection of DOD based on the regression model is proposed. There are some similarities between the projections and the corrected projections.*

*The paper is very interesting, includes novelties and deserves publication. However, I have several doubts and comments that need clarification and further discussion.*

*General comments*
*1) DOD from MODIS: It is not clear what the DOD derived from MODIS refers to. Is it total dust optical depth or coarse dust optical depth? I understand that it refers to the total dust optical depth (fine and coarse) but I was confused when the product was compared to the coarse (O'Neill) product from AERONET. Can you please explain better the derivation of DOD from AOD in the paper? Given the importance of the dataset for the paper I feel it is not enough to refer the reader to other publications. Also, can you provide an estimation of the uncertainty of this product?*

We added lines 102-114 to better explain how DOD is derived. It is coarse dust optical depth.  The formula is derived from the work of Anderson et al. (2005). Uncertainty of this product is added to the supplementary information as shown below. We also modified lines 119-122 to include these information.

Figures R1-2 compares aerosol optical depth (AOD) between MODIS and AErosol RObotic NETwork (AERONET) sites data (top), and between MODIS DOD and AERONET coarse mode aerosol optical depth (COD; bottom). AERONET COD is processed by the Spectral Deconvolution Algorithm (O'Neill et al., 2003). We used an evaluation method following Levy et al. (2003; their Fig. 11) for AOD and COD errors. The AERONET Level 2 (quality assured) 10 minutes AOD and COD (500 nm) are extracted for Aqua equatorial crossing time (1:30 PM) and Terra equatorial crossing time (10:30 AM) plus or minus 30 minutes, and are considered if there is at least 2

measurements per day and there should be at least 100 days with data. We select AERONET sites within a spatial radius of 15 km of MODIS measurement. 883 AERONET sites are used. Total number of valid data is about 35,747. In box-whisker plots (e.g., Fig. R1), all collocated MODIS and AERONET data are grouped into bins of 500 measurements. The last bin will contain a larger number of values corresponding to the remaining of the division.

As shown in Fig. R1, MODIS slightly underestimated Aqua AOD and DOD for most of the AOD and DOD ranges. Compared to AERONET station data, Aqua AOD is underestimated, and DOD largely inherits this error. For Aqua DOD around 0.50, the median error is around 0.08, with estimated errors ranging from -0.29 to 0.16. Terra DOD is better than Aqua DOD in terms of the median of errors (Fig. R2 bottom vs. Fig R1 bottom). The median error for Terra DOD around 0.50 is very close to zero, with estimated errors ranging from -0.23 to 0.25.

[Figure]

Figure R1. Comparison between grouped Aqua AOD error (i.e., the differences between MODIS AOD and AERONET AOD versus AERONET AOD, top), and grouped coarse mode aerosol optical depth (COD) error (i.e., the differences between MODIS DOD and AERONET COD versus AERONET COD, bottom). For each box-whisker, its width is 1σ of the AOD (COD) bin, while its height, whiskers, middle line and red dots are the 1σ, 2σ, mean, and median of AOD (COD) error, respectively. The envelope of estimated errors are blue and the one-one line (zero error) is dashed black.

[Figure]

Figure R2. Same as Fig. R1 but for Terra DOD.

*The confidence of satellite data over the different regions is assessed by comparison with AERONET (few stations, low spatial coverage), CALIOP, and considering the number of days with available DOD per season. Results show that while in Africa, South America, Middle East and some Asian regions confidence seems to be high, for some regions in*

*Asia/North America it largely depends on the season. In my view, the strength or confidence on the DOD data by region should be considered when discussing: the modelled DOD evaluation at the regional level, the regression method projections and discrepancies with CMIP5 models.*

Major uncertainties we found in terms of days of coverage and comparison with AERONET and CALIOP are: 1) low coverage over northern China and Southeastern Asia in JJA; 2) DOD is slightly higher than COD from AERONET over Arabian Peninsula in DJF and SON; 3) DOD is lower than CALIOP COD over northern India in MAM. We added lines 159-164, 338-343 to discuss the uncertainties associated with DOD.

*2) DOD from CMIP5 models: The authors compare the DOD derived from the selected CMIP5 models using Eq. (2). Using a value of 0.6 everywhere and for every model is an important simplification as it depends on model-dependent assumptions on size distribution and other issues such as the size range considered. While 0.6 may be a reasonable value for GFDL-CM3, how can we be sure it is ok for other models? Is there any other model for which you could compare this assumption in addition to GFDL-CM3.*

We agree that using 0.6 for all models is a simplification and adds uncertainties to our analysis. We modified text to address this issue, e.g., added lines 237-241: "Applying the same mass extinction efficiency everywhere and to all the CMIP5 model output used here is a simplification, as different models may have quite different mass extinction efficiency. For instance, $e$ can range from 0.25 to 1.28 $m^2$ $g^{-1}$ in AEROCOM models, with a multi-model medium of 0.72 $m^2$ $g^{-1}$ (Huneeus et al., 2011)." and lines 243-244: "A full validation of this method will require modeled DOD from all the other CMIP5 models, which are currently not available."

*3) Clear sky vs all sky values: While the authors have made an effort to gather the largest possible amount of DOD data by using both Aqua and Terra, the results of the comparison between MODIS DOD and model DOD may be quite affected by the use of all sky values from the models instead of clear sky values. Can you at least quantify this effect by for example using clear sky DOD from GFDL-CM3? How large is this effect? This may be potentially important in areas with seasonal clouds and precipitation. Could this be one of the reasons for the strong disagreement in some regions?*

As the reviewer pointed out, MODIS AOD removed pixels contaminated by cloud, and therefore AOD (and DOD) is retrieved toward a clear-sky condition. On the other hand, the derived (or modeled) DOD in CMIP5 models does not have any cloud-screening process and therefore is under an all-sky condition. The inconsistence between the two may add some uncertainties in regions with more cloud coverage/amount, such as the central U.S., northern China, southeastern Asia, and northern South America, but less so over North Africa, South Africa, the middle East, Australia, India (except JJA), and Australia (Figure R3).

**ISCCP Cloud Amount (1991-2012)**

[Figure]

Figure R3. Total cloud amount (%) from the International Satellite Cloud Climatology Project (ISCCP) averaged over 1991-2012. Black boxes denote the nine averaging regions.

In GFDL-CM3 model, DOD at each grid point is calculated under all-sky condition and model does not have output of clear-sky DOD. We compared DOD from CALIOP level 3 data under all-sky condition and cloud-free (i.e., clear sky) condition (Figure R4). The differences of global mean DOD over land under all-sky and clear-sky conditions range from -0.003 in MAM to 0.001 in DJF. The differences are larger (> ±0.05) over cloudy regions in MAM and JJA, particularly over Guinea coast in West Africa, northern China, southeastern Asia, India (Fig. R4, bottom). The differences are largely due to the fact that much less samples are collected to produce cloud-free DOD over these cloudy regions (not shown). The disagreement between MODIS DOD and CMIP5 DOD in the above regions (i.e., Guinea coast in West Africa, northern China, southeastern Asia, India) is not particularly higher than other regions (e.g., Fig. 4).

**CALIOP DOD 2007-2016 (all sky)**

[Figure]

**CALIOP DOD (all-sky minus cloud-free)**

[Figure]

Figure R4. Climatology (2007-2016) of CALIOP DOD under all-sky condition (top) and the differences between all-sky and cloud-free conditions (bottom). Blue numbers denote global mean DOD over land.

*4) Interannual variability: One of the findings of this study is that the interannual DOD variation is not very well captured by the CMIP5 models. It is stated that "models probably misrepresented these [controlling factor] relationships, in addition to their incapacity of capturing the interannual variations of individual controlling factors". Because of their nature, CMIP5 models cannot (and are not meant to) represent year-to-year variations of the driving factors in such a short time period. Therefore, the first part of the statement is just speculative, i.e., one cannot know whether the relationships are misrepresented from that analysis alone. I strongly believe that this part should be better discussed both in the results section and the conclusions. I also believe that the*

*comparison between CMIP5 model output and observations in Figures S4 to S6 is not needed. Isn't it obvious that CMIP5 models cannot represent year-to-year variations of each season in a 12-year period?*

The reviewer questioned our argument "...models probably misrepresented these relationships..." following the discussion on Figure 5 and Table S1. We did not intend to make any conclusion at that line, but to bring up a question. Later in Figure 6 we examined the connections between CMIP5 DOD and controlling factors. We revised line 425 to avoid misunderstanding: "... models may misrepresent these relationships, in addition to their incapacity of capturing the interannual variations of individual controlling factors in general". We also followed reviewer's suggestion to remove Figures S4-6 and modified text accordingly (line 427, lines 669-674).

*5) The role of surface bareness: one of the important conclusions of the study is that "constraints from surface bareness are largely underestimated while the influences of surface wind and precipitations are overestimated". I have a few doubts/comments on this:*
*a. How can you know that the constraint from surface bareness is largely underestimated? Given your methodology, couldn't it be that the constraint of surface bareness is correct in absolute terms but the effect of precipitation (through soil humidity) is overestimated? This should be clarified.*

It is possible that the magnitude of one controlling factor in the model is closer to the observation while the others are systematically underestimated/overestimated. So we standardized each controlling factor before regression. Therefore, the differences due to their absolute values are removed. The regression coefficients thus reflect how the interannual variations of each factor may contribute to the variations of DOD.

*b. While I think that the methodology is sound, it is not clear to me how year-to-year variations of around 2-3 % in LAI (Figure S7) can have such an impact in the interannual variability of dust in Northern Africa. Because this conclusion has important implications, could you further discuss this point? What would be the physical mechanisms that could explain this?*

First of all, we'd like to clarify that Fig. S7 shows bareness instead of LAI. Year-to-year variations of LAI are above 10% over the Sahel and parts of North Africa (Figure R5, right column). Bareness, or LAI, is a key non-erodible factor that can prevent wind erosion. The reason bareness shows a stronger influence on the interannual variations of DOD than the other two factors (precipitation and surface wind speed) is because its variations are more consistent with DOD changes. Here we show an example. We select an area over the Sahel (10°-16°N, 0°-25°E) where bareness is the dominant controlling factor in MAM based on multiple liner regression (Figure R6a). As shown in Figure R6b, the interannual variation of bareness (orange) is more consistent with DOD (black) variations than surface wind speed (green) or precipitation (purple) in the region. The correlation between DOD and standardized bareness is 0.61 (p=0.03), also higher than the correlations between DOD and precipitation (-0.46) or between DOD and surface wind (0.55).

We also examined multiple-linear regression using LAI from GLASS during 2004-2014 (Xiao et al. 2014). GLASS LAI is derived from MODIS products for years after 2001. The results using GLASS LAI are very similar to what we obtained from AVHRR LAI (Figure R7).

[Figure]

Figure R5. Seasonal mean of LAI averaged over 2004-2016 ($m^2/m^2$; left) and ratio (%) of standard deviation of LAI to seasonal mean LAI (right) over North Africa and the Arabian Peninsula from AVHRR during 2004-2016.

[Figure]

Figure R6. (a) Same as Fig. 6b but on a 1° by 1° grid for North Africa and the Middle East. Black box indicate an averaging area between 10°-16°N and 0°-25°E. (b) Time series of standardized controlling factors of bareness (orange), surface wind (green), precipitation (purple) and MODIS DOD (black) averaged over the area shown in (a).

[Figure]

Figure R7. Regression coefficients calculated by regressing DOD in each season onto standardized precipitation (purple), bareness (orange), and surface wind speed (green) from 2004 to 2014. Coefficients obtained using MODIS DOD and observed controlling factors. Plots in the left used LAI from the GLASS, while on the right used LAI from the AVHRR. All the other variables are the same. The color of the shading denotes the largest coefficient in absolute value among the three, while the saturation of the color shows the magnitude of the coefficient (from 0 to 0.04). All regression coefficients regardless of their statistical significance are shown. Missing values are shaded in grey. To highlight coefficients near dust source regions, a mask of LAI ≤ 0.5 is applied.

*Can you provide the same figure (S7) but for the model derived bareness (both present day and future projections)? How well do the models compare with the observed range of variability of the LAI in arid regions (the Sahara for example)?*

Modeled climatology of bareness is higher over North Africa (Figure R8) than that in the AVHRR, and the standard deviation is lower over northern North Africa but much higher over the Sahel.

[Figure]

Figure R8. Seasonal mean (left) and standard deviation (right) of bareness over North Africa and the Arabian Peninsula from CMIP5 7-model ensemble mean during 2004-2016.

*6) Regression method projections vs. CMIP5 projections: The regression method used to derive DOD in future scenarios is based upon 16 CMIP5 model variables (surface wind speed, bareness and precipitation) and compared to dynamical projections of only 7 CMIP5 models (those with online dust schemes). Partly, differences in future trends might come by differences in driving variables. You state [line 439] that projected DOD changes using the full sample or only 7 models are very similar. If so, why not using the same 7 model outputs as drivers? This would enhance consistency. Finally, why the similarities between the two approaches in some regions may be informative?*

The purpose of using variables from 16 CMIP5 models for the future projection is to include as much information (i.e., more model output) about projected change of the controlling factors as possible. As the reviewer pointed that, different number of CMIP5 models used for the regression model may add to the differences between CMIP5 model projected DOD and regression model projected DOD. So we follow the suggestion to show regression model projected DOD change using 7-model output in Figure 9 to keep the consistency and show results from 16-model in Figure S7.

We removed "which may be informative", and modified the sentence to: "we find some similarities between the two, which adds to the confidence of projected DOD change in these regions, for instance...". We also modified lines 513-522 to better explain approach of future projection using the regression model: "Here we also present the projected change of DOD from the regression model in Figure 9. The regression model is developed based on observed relationships between DOD and local controlling factors and can largely capture the interannual variations of DOD in the present-day climate (Table S1 in the Supplement). Assuming that the observed connection between DOD and these controlling factors do not change dramatically in the future, we can use this regression model and CMIP5-model projected change of controlling factors to project DOD variations. Compared to DOD projection from CMIP5 models, this approach utilizes additionally observational constrains and is likely to provide a more reliable future projection." Although CMIP5 models overestimate the influence of surface wind and precipitation and underestimate the role of bareness, there are some similarities between model and observations (Fig. 6; lines 450-462), which indicate that models partially capture the connection between the DOD and these controlling factors in some regions. Therefore, the projection of DOD from CMIP5 models is not completely unreliable. The similarity between CMIP5 projection and the projection from the regression model thus adds to the confidence of projected change of DOD over North Africa, the Arabian Peninsula, and northern China in some seasons.

*Minor comments:*
*- I suggest to list multiple references to the same topic chronologically, unless there are reason to order them differently, e.g. in the introduction.*
        Done.

*- I think the column heading "Dust emission scheme" is somewhat misleading as the given references describe the implementation of a dust emission scheme, not the scheme itself. Perhaps rewording to "Dust emission implementation" or similar would help.*
        We follow the suggestion to change column head to "Dust emission implementation".

*- I suggest changing Eq. (1) to Bareness = exp(-LAI). Also, is there a reference for this equation?*
        We change Eq. (1) following the comment and added a reference.

*- L. 146-147: I would normally not consider a resolution of 80km "very suitable to study the influence of wind speed on dust emission and transport on small scales". I understand the intent of this statement, but I suggest rephrasing this to avoid misunderstanding.*
        Thanks for the suggestion. We modified lines 185 to "We choose this analysis because of its relatively high spatial resolution".

*- L. 160: I suggest to delete "relatively high" as well as "quite"*
Done.

*- L. 192: GFLD-CM3 should be GFDL-CM3*
Done.

*- Line 205: clarify which DOD is regressed onto observed values, i.e. satellite derived DOD*
We changed the sentence to "by regressing MODIS DOD onto..."

*- Fig. 3i: It is very hard to see the MODIS DOD pattern for Australia. Can this be improved?*
We re-plotted the figure to better display MODIS DOD for Australia.

*- Fig. 4 is ok, but quite dense*
We updated the figure to make it look better.

*- L. 311: variability instead of variations*
Done.

*- L. 328: wind erosion instead of "soil erosion from wind"*
Done (now line 173).

*- Line 431: centaury should be century*
Done.

*- L. 457 ff: Sometimes it is not clear if "models" refers to the CMIP5 models or projection 'models'.*
We changed "models" to "regression projections" to avoid confusion.

*- Figure 6. It is difficult to sort out the different elements, e.g. the strength of the regression depending on the shading intensity is not visible. I would suggest: to make a zoom per region, or to display dependencies from the 3 variables in independent maps, and to use the same resolution for MODIS and CMIP5 maps to make easier a direct comparison.*
We updated Figures 6 by interpolating results from MOIDS and observed controlling factors to model grids (2° by 2.5°, Figs. 6a-d) and changed color scale from 0~0.02 to 0~0.04 to better show the shading intensity. We also zoomed in and plotted a few figures for different regions (Figures R9-R13) here. The patterns in new Fig. 6 are very similar to the old one. The connection between DOD and bareness is underestimated on the interannual time scale in CMIP5 models. On the other hand, DOD's connection with precipitation and surface wind speed are overestimated.

[Figure]

Figure R9. Same as Fig. 6 but for Africa and the Middle East. Black boxes denote the averaging regions defined in Table 2: North Africa, South Africa, and the Middle East.

[Figure]

Figure R10. Same as Fig. 6 but for Asia. Black boxes denote the averaging regions defined in Table 2: northern China, India, and southeastern Asia.

[Figure]

Figure R11. Same as Fig. 6 but for North America. Black boxes denote the averaging regions defined in Table 2: North America.

[Figure]

Figure R12. Same as Fig. 6 but for Australia.

[Figure]

Figure R13. Same as Fig. 6 but for South America. Black boxes denote the averaging regions defined in Table 2: South America.

Reference:

Xiao ZQ, Liang SL, Wang JD, Chen P, Yin XJ, Zhang LQ, et al. Use of general regression neural networks for generating the GLASS leaf area index product from time-series MODIS surface reflectance. IEEE T. Geosci. Remote 52, 209-223 (2014)

[revised manuscript text omitted]